# HMFGraph: Novel Bayesian approach for recovering biological networks

**Aapo E. Korhonen***, **Olli Sarala, Tuomas Hautamäki**, **Markku Kuismin**,
**Mikko J. Sillanpää***

Research Unit of Mathematical Sciences, University of Oulu, Oulu, Finland

* aapo.korhonen@oulu.fi (AEK); mikko.sillanpaa@oulu.fi (MJS)

**Data availability statement:** All codes for reproducing the results are publicly available at

## Abstract

Gaussian graphical models (GGM) are powerful tools to examine partial correlation structures in high-dimensional omics datasets. Partial correlation networks can explain complex relationships between genes or other biological variables. Bayesian implementations of GGMs have recently received more attention. Usually, the most demanding parts of GGM implementations are: (i) hyperparameter tuning, (ii) edge selection, (iii) scalability for large datasets, and (iv) the prior choice for Bayesian GGM.

To address these limitations, we introduce a novel Bayesian GGM using a hierarchical matrix-F prior with a fast implementation. We show, with extensive simulations and biological example analyses, that this prior has competitive network recovery capabilities compared to state-of-the-art approaches and good properties for recovering meaningful networks. We present a new way of tuning the shrinkage hyperparameter by constraining the condition number of the estimated precision matrix. For edge selection, we propose using approximated credible intervals (CI) whose width is controlled by the false discovery rate. An optimal CI is selected by maximizing an estimated $F_1$-score via permutations. In addition, a specific choice of hyperparameter can make the proposed prior better suited for clustering and community detection. Our method, with a generalized expectation-maximization algorithm, computationally outperforms existing Bayesian GGM approaches that use Markov chain Monte Carlo algorithms.

The method is implemented in the R package `HMFGraph`, found on GitHub at https://github.com/AapoKorhonen/HMFGraph. All codes to reproduce the results are found on GitHub at https://github.com/AapoKorhonen/HMFGraph-Supplementary.

## Author summary

In this paper, we introduced a new way to recover network structures with biological datasets in mind. Network estimation methods can help research by providing a convenient and easy-to-understand way to explain multivariate data structures and variable interactions. This can show, for example, how different genes co-express. Here, we

https://github.com/AapoKorhonen/
HMFGraph-Supplementary. The showcased
method can be tested with our R package
HMFGraph, also publicly available at Github:
https://github.com/AapoKorhonen/HMFGraph.

**Funding:** This research was supported by the
Research Council of Finland (grants 349393,
329439), the Finnish Doctoral Program
Network in Artificial Intelligence, AI-DOC
(decision number VN/3137/2024-OKM-6), and
Vilho, Yrjö and Kalle Väisälä Foundation. The
funders had no role in study design, data
collection and analysis, decision to publish, or
preparation of the manuscript.

**Competing interests:** The authors have
declared that no competing interests exist.

introduced a new model structure that has competitive network recovery capabilities
compared to state-of-the-art methods in a wide range of simulation settings. In addition,
we include examples with real datasets and explain how the interpretation changes with
different model parameter values. Estimation is performed by using our fast algorithm,
which has significant computational advances over conventional estimation methods.
Our method has a user-friendly implementation as an R package and is publicly available
for download on GitHub at https://github.com/AapoKorhonen/HMFGraph.

## 1. Introduction

Data structures and conditional dependencies between multiple variables can be easily
explored with graphical models. Gaussian graphical models (GGM) are used for network
estimation, and they provide a convenient way of producing partial correlation networks [1].
They have been studied for a long time, and a wide range of methods have been developed,
while recently Bayesian methods are increasingly gaining popularity [2,3].

Biological networks estimated with GGM can reveal complex associations between data
variables, e.g., finding gene co-expression patterns in gene expression datasets [4,5]. GGMs
are general tools that can be applied to various biological datasets, inferring, e.g., cancer path-
ways, metabolic networks, protein networks, gene networks, or microbiome networks [6–8].
GGMs provide a convenient and easy-to-interpret representation, which may help researchers
to make novel biological discoveries. This includes finding new connections between genes,
operational taxonomic units (OTUs), or other omics variables, as well as identifying groups
of genes that are potentially operating in related biological processes by inspecting cluster-
ing structures between them [9]. In addition, biological networks can help identify important
cancer genes [10]. Biological datasets often have properties that must be taken into account
during the network estimation procedure. For example, they may be high-dimensional, and
the networks could exhibit scale-free or cluster-like structures.

With GGMs, the data are assumed to follow a multivariate normal distribution, and the
partial correlation structure is constructed from an estimated inverse covariance (precision)
matrix. Non-zero elements of the precision matrix correspond to non-negligible partial cor-
relations, which again determine the edges of the graph [11]. In high-dimensional situations,
i.e., when there are fewer samples than variables, we need to introduce regularization to the
problem in order to produce well-conditioned estimates for covariance and precision matri-
ces. The most notable frequentist method for GGM is the graphical lasso (Glasso), which uses
a lasso penalty to induce sparsity in the estimated precision matrix.

When estimating covariance and precision matrices in the Bayesian framework, we have
to use a prior that ensures positive definite estimates and regularizes the estimator. Wishart
and inverse-Wishart prior distributions are suitable priors for precision and covariance matri-
ces, respectively, since they fulfil both of these criteria. Wishart distribution is also a conjugate
prior for (multivariate) normally distributed data, giving a closed-form expression for the cor-
responding posterior distribution. Thus, Wishart and inverse-Wishart priors have been used
in a wide range of applications for covariance and precision matrix estimation [12]. There are
some extensions for Wishart prior because it can be too inflexible due to only having a single
tuning parameter [13]. These include hierarchical Wishart [14], matrix-F (scale mixture of
Wisharts) [15], and flexible inverse-Wishart prior [13]. This paper aims to extend this line of
research further by providing a hierarchical version of the matrix-F prior.

Previously, Wishart prior has been widely used for Bayesian GGMs. Wishart prior introduces a ridge-type regularization to the precision matrix, which will not yield a sparse estimate. If sparsity is preferred, an additional post-selection step (via decision rule) is required to determine which elements can be set to zero. Previously, a couple of decision rules have been used in this context, including the extended Bayesian criterion [16], Bayes factors [17], and posterior probabilities (credible intervals) [18]. In [17], they used an empirical Bayesian method for selecting an optimal hyperparameter for the Wishart prior and analytically calculated values for the Bayes factor to effectively select edges for the graph. This method turns out to be computationally extremely efficient. On the other hand, it has the same possible problems as other Wishart prior models, namely inflexibility.

An extension for Wishart distribution, namely G-Wishart, was developed with graphical models in mind. A graph estimate is produced at each Markov chain Monte Carlo (MCMC) step. A major drawback of G-Wishart is that it necessitates using complicated sampling algorithms. There is a direct sampler for G-Wishart [19], but it is computationally demanding for large graphs.

False discovery rate (FDR) control has previously been used with network estimation methods [17,20,21]. By controlling the target FDR, we can set an a priori expectation for the number of wrong edges. For instance, if we set our target FDR to 0.2, then we can expect 20% of our estimated network's edges to be incorrect. In some cases, we have to tolerate some level of false positives in order to estimate any network at all. For example, a network estimate that is too sparse can make clusters difficult to recognize. By managing the FDR, we can gain a clear understanding of the extent to which we need to tolerate false edges.

In Bayesian analysis, the posterior distribution can be estimated by generating dependent samples from the posterior density. This can be done with MCMC methods [22] and one of the simplest options is the Gibbs sampler [23]. It is usable in cases where a posterior distribution is not analytically available, but the full conditional distribution for each parameter can be derived. Gibbs sampler is easily available with conjugate priors. By using MCMC methods, the maximum a posteriori (MAP) estimate and the shape of the posterior density (summarizing uncertainty) are attained. In [24] and [15], a Gibbs sampler for matrix-F prior was introduced.

The primary novelties in this paper are:

- A novel hierarchical matrix-F prior for Bayesian GGM with a user-friendly and flexible parameterization.
- A generalized expectation-maximization (GEM) algorithm that has remarkable computational advantages over previous MCMC methods.
- A new way of tuning the hyperparameter by constraining the condition number of the estimated precision matrix.
- Credible intervals (CI) are approximated with a normal distribution.
- The optimal CI is selected with permutations and maximizing an estimated $F_1$-score. This also allows us to control the FDR to a desired level.

We show that our method works well with high-dimensional data and performs considerably better than current state-of-the-art methods. It is flexible and has good network recovery performance in all tested scenarios. Finally, we include examples with real biological datasets and show how a specific choice of hyperparameter can make the proposed prior better suited for clustering and community detection.

The rest of this paper is structured as follows. In Sect 2, we introduce the method in detail. In Sects 3.1 and 3.2, we provide examples and results with simulated and real datasets, respectively. In Sect 4, we discuss the results, the method, and possible future improvements.

## 2. Methods and implementations

In this section, we will describe our approach in detail. Each major step with the approach is showcased in our flowchart in Fig 1. All important mathematical symbols and notation are explained in Table 1.

### 2.1. Background on Gaussian graphical models

The data $Y$ is expected to follow a multivariate normal distribution in GGMs with a zero-mean vector and a covariance $\Sigma = \Omega^{-1}$, and $\Omega$ being the precision matrix or inverse of the symmetric and positive-definite covariance matrix. For all samples $j$: $Y_j \sim N(\mu, \Omega^{-1})$, where $\mu = (0, 0, \ldots, 0)^\top$ is $p$-length zero vector, where $p$ is the number of variables.

Graphs $G = (V, E)$ are represented as a set of vertices $V$ and edges $E$ [11]. If $(ij) \in E$, then vertices $v_i$ and $v_j$ are connected in the graph. With GGMs, connections in the network are determined by off-diagonal elements of the precision matrix $\Omega = [\omega_{ij}]$. If $\omega_{ij} \neq 0$ then $(ij) \in E$.

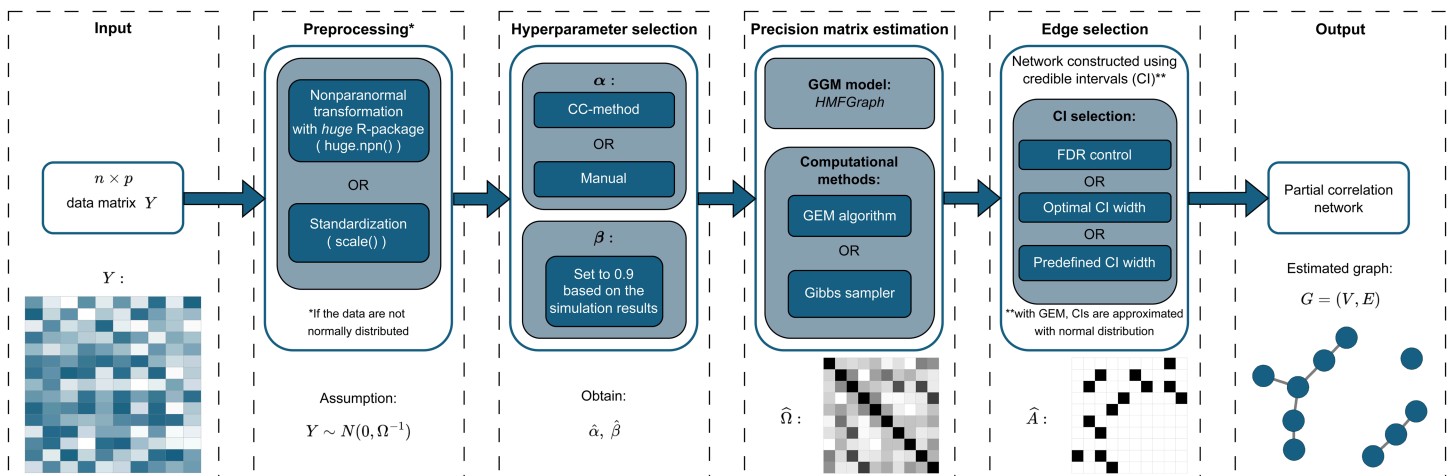

**Fig 1. Flowchart of the proposed Bayesian approach for network recovery.**

**Table 1**. **Mathematical symbols and notation used throughout this article.**

| Symbol | Description |
|---|---|
| $\Omega$ | A precision matrix (inverse of the covariance matrix) |
| $\Phi$ | A model parameter: the target matrix of $\Omega$ |
| $B$ | A model parameter: the target matrix of $\Phi$ |
| $\nu, \delta$ | Hyperparameters of the matrix-F distribution |
| $\alpha$ | A transformed model hyperparameter from $\nu$: controls the shrinkage of the diagonal elements of the precision matrix |
| $\beta$ | A transformed model hyperparameter from $\delta$ and $\nu$: controls the shrinkage of the off-diagonal elements of the precision matrix |
| $\gamma$ | A probability mass of equal-tailed credible interval |
| $A$ | An adjacency matrix |

This information is compiled in a symmetric adjacency matrix $A = [a_{ij}]$, which is defined as:

$$a_{ij} = \begin{cases} 1, & \text{if } \omega_{ij} \neq 0 \\ 0, & \text{if } \omega_{ij} = 0. \end{cases} \tag{1}$$

## 2.2. Bayesian Gaussian graphical models and matrix-F prior

Posterior distribution in the Bayesian framework is obtained from the likelihood and prior distribution:

$$p(\Omega|Y) \propto p(Y|\Omega)p(\Omega), \tag{2}$$

where $p(Y|\Omega)$ is a likelihood and $p(\Omega)$ is a prior. The likelihood function in Bayesian GGM is

$$p(Y|\Omega) \propto |\Omega|^{n/2} \exp\left[-\frac{1}{2}\text{tr}(nS\Omega)\right], \tag{3}$$

where $S$ is a sample covariance matrix. Here $\Omega$ is a positive-definite matrix.

The matrix-F distribution was introduced in [15] as a suitable and flexible prior for covariance and precision matrices. In contrast to one parameter for the degree of freedom in the regular Wishart distribution, the matrix-F has two parameters. The matrix-F prior was used in the Bayesian GGM setting by [24]. The prior is

$$\Omega \sim F(B^{-1}, \nu, \delta), \tag{4}$$

where $B$ is a target matrix, $\nu > p - 1$ and $\delta > 0$ are degree of freedom parameters. The parameter $\nu$ controls the center of the distribution, and the parameter $\delta$ controls the tail behavior.

For Bayesian GGM, the matrix-F can be represented as a scale mixture of Wisharts:

$$F(B^{-1}, \nu, \delta) = \int W(\Omega; \nu, \Phi^{-1}) W(\Phi; \delta + p - 1, B^{-1}) d\Phi, \tag{5}$$

and it can be modeled with two separate priors:

$$\Omega \sim W(\nu, \Phi^{-1}), \tag{6}$$
$$\Phi \sim W(\delta + p - 1, B^{-1}), \tag{7}$$

where $\Phi$ is a target matrix for the first Wishart distribution, and it can be thought of as an auxiliary parameter.

We propose a modification to the matrix-F prior so that the parameters are simpler to interpret. We first set the same Wishart scale mixture as in (6) and (7), but we add scaling terms $(\nu - p - 1)$ and $(\delta + p - 1)$ for target matrices $\Phi$ and $B$, respectively. This modification will simplify the interpretation of the model hyperparameters.

We set the parameter $B$ to be a diagonal matrix, i.e., all off-diagonal elements are zero. For diagonal elements $b_{ii}$, a flat prior on a logarithmic scale was proposed to be used with the Wishart prior in [25]. [14] implemented it with a Metropolis-Hasting-within-Gibbs algorithm. Interestingly, this prior is similar to $\text{Gamma}(\text{shape} = \epsilon_1, \text{rate} = \epsilon_2)$ with small values of $\epsilon_1$ and $\epsilon_2$ (e.g., 0.001). In fact, when $\epsilon_1, \epsilon_2 \to 0$, then $b_{ii} \sim \text{Gamma}(\epsilon_1, \epsilon_2) \approx 1/b_{ii}$.

The gamma prior ensures a proper posterior distribution, but the flat prior on the logarithmic scale is scale-invariant and should be more flexible when working with differently scaled data. In Section M in S1 Text, we include proof that the posterior is proper with the improper

prior when $n > p$. The posterior is proper with the gamma prior even when $n < p$ (see Section N in S1 Text). Therefore, we use the gamma prior for the diagonal elements of the matrix $B$.

## 2.3. Hierarchical matrix-F prior for Gaussian graphical models

Our model, using the scaled matrix-F prior and the gamma prior, is

$$Y_j | \Omega \sim N\left(0, \Omega^{-1}\right), \; j = 1, 2, \dots, n, \tag{8}$$

$$\Omega | \Phi \sim W\left(\nu, \left((\nu - p - 1)\Phi\right)^{-1}\right), \tag{9}$$

$$\Phi | B \sim W\left(\delta + p - 1, \left((\delta + p - 1)B\right)^{-1}\right), \tag{10}$$

$$b_{ii} \sim \text{Gamma}(\epsilon_1 = 0.001, \epsilon_2 = 0.001), \; i = 1, 2, \dots, p. \tag{11}$$

Hereafter, we call this model formulation a *hierarchical matrix-F prior for Gaussian graphical models*, or HMFGraph.

A significant benefit of these priors is that the full conditional distributions are known for all parameters $\Omega$, $\Phi$, and $B$ (Section A in S1 Text). The modes of the full conditional distributions are used in our GEM algorithm. For parameters $\Omega$ and $\Phi$, they are

$$\text{Mode}(\Omega | \cdot) = (\nu + n - p - 1)\left((\nu - p - 1)\Phi + nS\right)^{-1}, \tag{12}$$

$$\text{Mode}(\Phi | \cdot) = (\delta + \nu - 2)\left((\delta + p - 1)B + (\nu - p - 1)\Omega\right)^{-1}, \tag{13}$$

where $\Omega | \cdot$ and $\Phi | \cdot$ are the full conditional distributions of the parameters $\Omega$ and $\Phi$, respectively.

Because of our scaling terms, we can present the modes of conditional distributions with new parameters, which are $\alpha$ and $\beta$:

$$\alpha = \frac{\nu - p - 1}{\nu + n - p - 1}, \quad \beta = \frac{\delta + p - 1}{\delta + \nu - 2}, \tag{14}$$

where $\alpha \in [0, 1[$ and $\beta \in [0, 1]$ (the lower limit of $\beta$ depends on $\nu$, see Section A in S1 Text). Similar substitution was done together with ordinary Wishart distribution in [17] and [26]. The modes of conditional distributions of parameters $\Omega$ and $\Phi$ utilized in our GEM algorithm can now be presented in an interpretable way:

$$\text{Mode}(\Omega | \cdot) = (\alpha\Phi + (1 - \alpha)S)^{-1}, \tag{15}$$

$$\text{Mode}(\Phi | \cdot) = (\beta B + (1 - \beta)\Omega)^{-1}. \tag{16}$$

The mode (15) bears a similarity to Ledoit-Wolf shrinkage (or a linear shrinkage) [27].

## 2.4. Computational methods for the posterior distribution estimation

We formulated two methods for estimating the posterior distribution of the HMFGraph model. The first is a Gibbs sampler that can estimate the MAP and the shape of the posterior distribution of the parameter $\Omega$. The second is a GEM algorithm that is computationally faster than the Gibbs sampler but is only able to acquire the MAP estimate [28,29]. We use the GEM algorithm in all example analyses unless specified otherwise.

The Gibbs sampler is explained in Section B in S1 Text. The GEM algorithm will be introduced in the next section.

**2.4.1. GEM algorithm.** The generalized expectation-maximization algorithm, or GEM algorithm, can be used in various statistical applications (e.g., [30]). Commonly, it is used for maximum likelihood and MAP estimation [28]. GEM and EM algorithms were originally designed for estimation problems with incomplete data or with hidden variables. These algorithms, in general, include two main steps: E-step, where the $Q$-function is updated with new hidden variables (or incomplete data), and M-step, where parameters are updated by maximization. The main difference between EM and GEM algorithms is the M-step. With the EM algorithm, the $Q$-function is maximized in terms of parameters in the M-step. With GEM, the $Q$-function does not need to be fully maximized in the M-step; it just needs to increase it (see [29, p. 24]). This makes the algorithm more usable with multiple parameters.

In our case, we operate in a complete data situation, thus the E-step becomes an identity operation [29]. This means that all M-steps in the algorithm are parameter-wise conditional maximizations of the posterior distribution. Since we already know the full conditional posterior distributions for each parameter, we can maximize each conditional distribution, and the algorithm turns into a so-called conditional maximization algorithm (see [29, p. 162]). In other words, the GEM algorithm is the same as the Gibbs sampler for our model, but instead of sampling from the distribution, we calculate its mode, which is usually analytically available. In fact, first forming the Gibbs sampler greatly helps to formulate a GEM algorithm [31]. The similarities between the Gibbs sampler and the GEM algorithm are most thoroughly examined in [32]. Also, the expectation conditional maximization algorithm, a special case of GEM algorithms, is analogous to the multi-step Gibbs sampler [29]. Our GEM algorithm is showcased in Algorithm 1 and in Section C in S1 Text. The algorithm can also be called the ECM or iterative conditional modes (ICM) algorithm [33]. It should be noted that for our model, a regular EM algorithm is not possible to implement, but the more flexible nature of the GEM algorithm allows for an easy implementation.

**Algorithm 1 GEM algorithm for Bayesian GGM using a hierarchical matrix-F prior.**

**Require:** max-iters, Stop-criterion, $\nu$, $\delta$, $n$, $p$
 $\alpha = (\nu - p - 1)/(\nu + n - p - 1)$
 $\beta = (\delta + p - 1)/(\delta + \nu - 2)$
 $B \leftarrow I_p$, $\Phi \leftarrow I_p$, $\Omega \leftarrow I_p$, $\Omega_{old} \leftarrow I_p$
 **for** $i \leftarrow 1$ to max-iters **do**
 **for** $i \leftarrow 1$ to $p$ **do**
 $\text{shape}_{ii} \leftarrow (\delta + p - 1)/2 + \epsilon_1$
 $\text{rate}_{ii} \leftarrow (\delta + p - 1)\phi_{ii}/2 + \epsilon_2$
 $b_{ii} \leftarrow (\text{shape}_{ii} - 1)/\text{rate}_{ii}$
 **end for**
 $\Phi \leftarrow (\beta B + (1 - \beta)\Omega)^{-1}$
 $\Omega \leftarrow (\alpha \Phi + (1 - \alpha)S)^{-1}$
 *relative F-norm* $\leftarrow ||\Omega - \Omega_{old}||_F/||\Omega_{old}||_F$
 **if** *relative F-norm* < Stop-criterion **then**
 **break**
 **end if**
 $\Omega_{old} \leftarrow \Omega$
 **end for**

## 2.5. Hyperparameter selection

With our model formulation, there are two hyperparameters that need to be tuned. They are $\alpha$ (or $\nu$) and $\beta$ (or $\delta$). Their minimum values correspond to the minimal information priors in the Bayesian framework [15].

The parameter $\alpha$ controls how much we trust the sample covariance matrix. In high-dimensional situations (i.e., $p > n$), this trust should be low, and $\alpha$ should then be set to a large value (close to one). As the sample size increases, the $\alpha$ should decrease. The exact value is still difficult to select by hand, especially for different data types, even though the interpretation is clear. We propose a method to select a suitable value for $\alpha$ based on a constraint on a condition number, which has been proven to construct well-conditioned estimators for covariance and precision matrices [16,34–36]. The condition number is the ratio of the largest and smallest eigenvalues of the estimated precision matrix:

$$\mathrm{Cond}(\widehat{\Omega}_\alpha) = \frac{\lambda_{\max}}{\lambda_{\min}}, \tag{17}$$

where $\widehat{\Omega}_\alpha$ is the MAP estimate of the precision matrix acquired with a some $\alpha$ value, and $\lambda_{\max}$ and $\lambda_{\min}$ are the largest and smallest eigenvalues of the precision matrix $\widehat{\Omega}_\alpha$, respectively.

Our proposed method chooses the smallest $\alpha$ value that produces a smaller condition number than the predetermined limit $\kappa_{\max}$:

$$\hat{\alpha} = \mathrm{argmin}_\alpha\left(\mathrm{Cond}(\widehat{\Omega}_\alpha) < \kappa_{\max}\right). \tag{18}$$

This raises a new problem: how to choose a correct $\kappa_{\max}$ value. Here, we set the constraint for the condition number to be lower than the condition number of a Ledoit-Wolf estimate (LW) of the same precision matrix [27,37]. The optimal $\alpha$ value is

$$\hat{\alpha} = \mathrm{argmin}_\alpha\left(\mathrm{Cond}\left(\widehat{\Omega}_\alpha\right) < \mathrm{Cond}\left(\widehat{\Omega}_{\mathrm{LW}}\right)\right), \tag{19}$$

where $\mathrm{Cond}\left(\widehat{\Omega}_\alpha\right)$ is the condition number for estimated $\Omega$ with some $\alpha$ value, and $\mathrm{Cond}\left(\widehat{\Omega}_{\mathrm{LW}}\right)$ is the condition number of a LW-estimate of the precision matrix. We call this a *condition number constraint* -method, or CC-method. We calculate the LW-estimate $\widehat{\Omega}_{\mathrm{LW}}$ using the R package `nlshrink` and the function `linshrink_cov`. We use a simple binary search algorithm (Algorithm A and Section E in S1 Text) to find a solution to Equation (19). In order to speed up the algorithm, we utilized a "warm-start" trick [38,39].

The parameter $\beta$ controls how much we trust the target matrix $B$. Because the target matrix $B$ is a diagonal matrix, a large $\beta$ value will induce a large regularization on off-diagonal elements and shrink them towards zero (see Fig 2). This shrinkage will also reduce the number of connections in the estimated network (see Fig G in S1 Text). The shrinkage is less noticeable if $\alpha$ is selected using the CC-method (see Figs F and G in S1 Text). This can be interpreted such that, if the true network is expected to be sparse, the $\beta$ value should be set to a large value. The larger the $\beta$ value, the sparser the obtained network. Based on our experience, the $\beta$ value does not affect the results as much as $\alpha$ due to the hierarchical structure of our prior and the CC-method (see Figs A-D in S1 Text). Similar behavior was observed with $\delta$ when using the matrix-F prior [24]. For all results, we used $\beta = 0.9$. We include more discussion on hyperparameter selection in Section D in S1 Text.

## 2.6. Edge selection with credible intervals

Because our model will not produce a sparse estimate for the precision matrix, we have to add an edge selection step after the estimation procedure. This is true for both the Gibbs sampler and the GEM algorithm. Note that this kind of edge selection is also required for models using sparsity-inducing priors, like spike-and-slab and horseshoe priors, that should produce

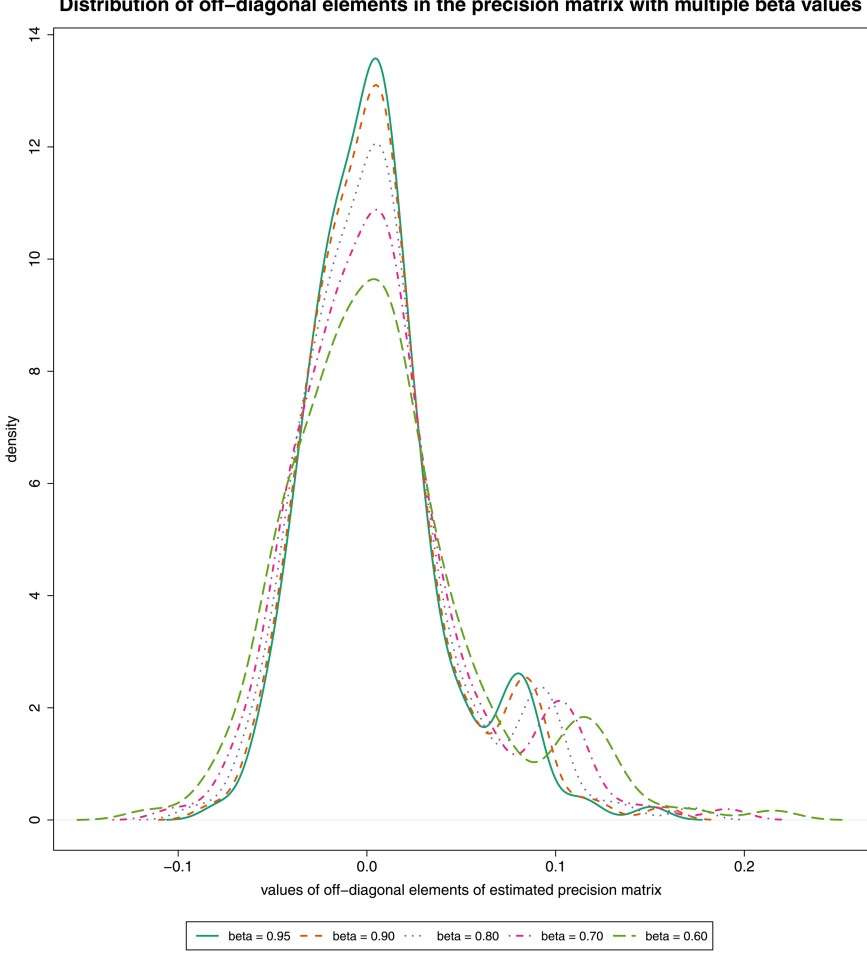

**Fig 2. Distribution of off-diagonal elements in the estimated precision matrix with hyperparameters: varying $\beta$ values (colors) and fixed $\alpha = 0.80$.** The graphical model considered in this example is a scale-free network generated with the R package `huge` ($n = 100$, $p = 20$).

a sparse precision matrix estimate (see e.g., [40–42]). Previous methods using Wishart priors have, for instance, used Bayes factors as a decision rule for edge selection [17,24]. Here, we use equal-tailed credible intervals for the edge selection.

The posterior distribution can be used to select nonzero elements from the estimated precision matrix by investigating if zero can be found from the credible interval. This can be expressed as:

$$\hat{a}_{ij} = \begin{cases} 1, & \text{if } 0 \notin \text{CI}(\gamma)_{\hat{\omega}_{ij}} \\ 0, & \text{if } 0 \in \text{CI}(\gamma)_{\hat{\omega}_{ij}}, \end{cases} \tag{20}$$

where $\text{CI}(\gamma)_{\hat{\omega}_{ij}}$ is ($ij$) element's $100 \cdot \gamma\%$ credible interval. For example, we could use a 95% credible interval ($\gamma = 0.95$) as a selection rule to construct our adjacency matrix. If value zero is not included in the 95% credible interval of an off-diagonal element of the precision matrix, the corresponding element will be set to 1 in the adjacency matrix $\widehat{A} = [\hat{a}_{ij}]$.

## 2.7. Approximation for credible intervals

When we estimate the MAP for the precision matrix with the GEM algorithm, we neither obtain an estimate for the posterior distributions nor credible intervals. As a solution for this, we chose to use a normal approximation for the posterior distribution of each off-diagonal element of the estimated precision matrix $\widehat{\Omega}$, as was suggested in [18]. Also, the variance-gamma distribution could be used [43] (see Section F in S1 Text). For the mean of the normal approximation, we use the MAP estimate. We know that the full conditional distribution for $\Omega$ is the Wishart distribution with known variance (Equations (G) and (P) in S1 Text). We can use this for the normal approximation. The approximation for (*ij*) off-diagonal element is

$$P\left(\omega_{ij}|Y\right) \approx N\left(\hat{\omega}_{ij}, \widehat{\mathrm{Var}}\left(\hat{\omega}_{ij}\right)\right), \tag{21}$$

where $\hat{\omega}_{ij}$ is the MAP estimate of (*ij*) off-diagonal element of the precision matrix and $\widehat{\mathrm{Var}}\left(\hat{\omega}_{ij}\right)$ is its estimated variance. The estimated variance of the MAP estimate is

$$\widehat{\mathrm{Var}}(\hat{\omega}_{ij}) = (\nu + n)\left(\hat{w}_{ij}^2 + \hat{w}_{ii}\hat{w}_{jj}\right), \tag{22}$$

where $\widehat{W} = \left[\hat{w}_{ij}\right] = (nS + (\nu - p - 1)\widehat{\Phi})^{-1}$ and $\widehat{\Phi}$ is a MAP estimate of the parameter $\Phi$ (Equations (G) and (P) in S1 Text).

Credible intervals for off-diagonal elements can now be estimated:

$$\mathrm{CI}(\gamma)_{\hat{\omega}_{ij}} \approx \left[\hat{\omega}_{ij} \pm \varphi^{-1}\left(\frac{1+\gamma}{2}\right)\sqrt{\widehat{\mathrm{Var}}\left(\hat{\omega}_{ij}\right)}\right], \tag{23}$$

where $\varphi^{-1}()$ is a probit function or the inverse cumulative distribution function for the standard normal distribution.

We compare these approximated credible intervals to the credible intervals that we obtain from the Gibbs sampler and the network recovered based on them in Figs 3 and 4. The GEM algorithm forms almost the same credible intervals as the Gibbs sampler, and this approximation can be considered useful and accurate for our purpose. The recovered networks based on CIs are similar.

## 2.8. False discovery rate control with permutations

The false discovery rate (FDR) describes the proportion of false positives among the estimated edges. FDR is generally defined as:

$$\mathrm{FDR} = \frac{\text{False positives (FP)}}{\text{True positives (TP)} + \text{False positives (FP)}}. \tag{24}$$

In multiple testing situations, it is necessary that we can control the FDR [44]. Network estimation can also be considered to suffer from multiple testing issues. With $p$ number of variables, we have to carry out $p \cdot (p-1)/2$ number of tests [45]. As $p$ increases, the number of tests increases greatly. Various ways to control FDR have been used with network estimation methods [17,20,21]. In order to do a frequentist-style multiple-testing correction in the Bayesian framework, we have to adjust the width of the credible interval.

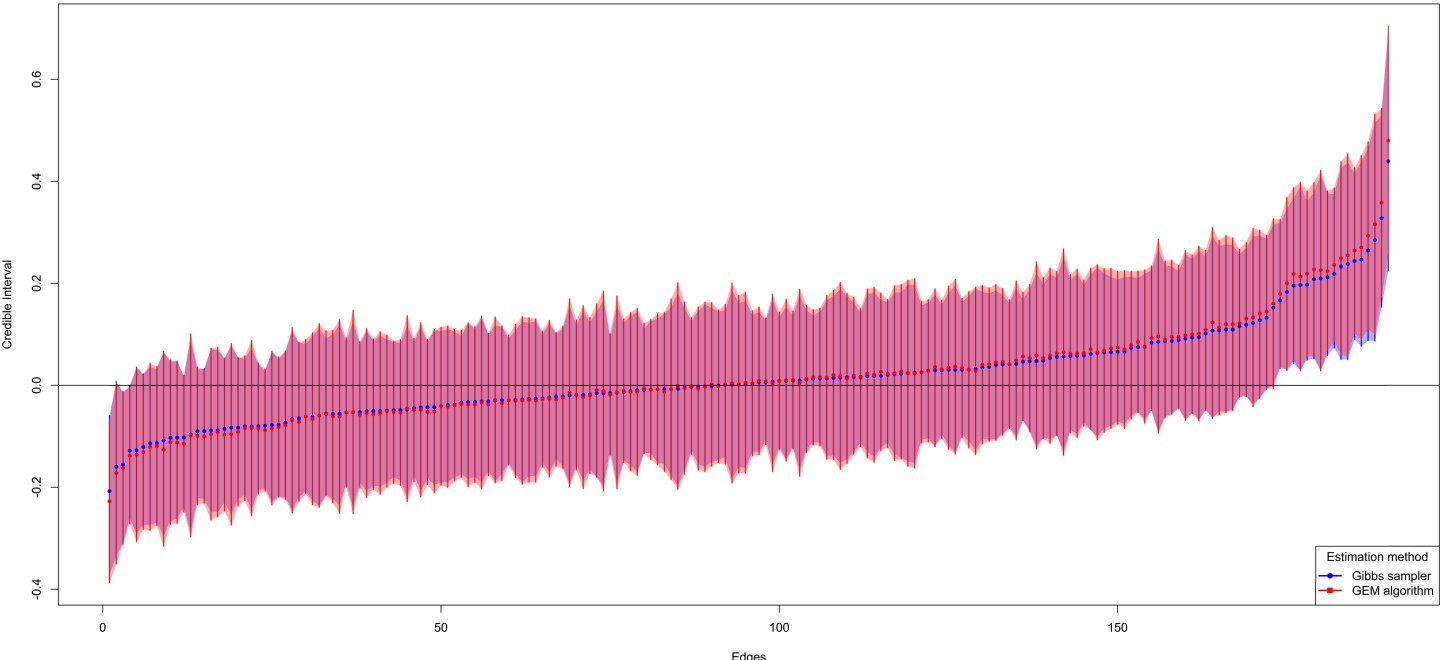

**Fig 3. Comparisons of 90% credible intervals (CIs) for off-diagonal elements of precision matrix $\Omega$.** For the Gibbs sampler, CIs are calculated from posterior samples. For the GEM algorithm, they are estimated with a normal approximation. The graphical model considered in this example is a scale-free network generated with the R package `huge` ($n = 100, p = 20$).

For networks, we define FDR as:

$$\text{FDR} = \frac{\sum_{(ij)\in\Theta} \hat{a}_{ij(\text{CI}(\gamma))}}{\sum_{i<j} \hat{a}_{ij(\text{CI}(\gamma))}}, \ (ij) \in \Theta \Leftrightarrow a_{ij} = 0, 1 \leq i < j \leq p, \tag{25}$$

where $\Theta$ includes all off-diagonal elements that are zeros in the hypothetical true adjacency matrix $A$. The true adjacency matrix $A$ is only acquirable with simulated data or under some specific hypothesis. The estimated adjacency matrix $\widehat{A}$ is dependent on the $\text{CI}(\gamma)$. The total number of true and false positives (denominator of Equation (24)) is simply the number of edges or nonzero elements in the upper or lower triangle of the estimated adjacency matrix ($\sum_{i<j} \hat{a}_{ij(\text{CI}(\gamma))}$). False positives (numerator of Equation (24)) are the number of edges found, of which true partial correlations are zero or negligibly small ($\sum_{(ij)\in\Theta} \hat{a}_{ij(\text{CI}(\gamma))}$).

The number of false positives is estimated using a permutation method. We expect that the permutation eliminates all correlation structures present in the original data. The permuted data will be obtained by permuting variable values within each sample separately, assuming the variables have a similar scale. For permuted data, all off-diagonal elements in the true adjacency matrix $A_P$ are hypothesized to be zeros ($\sum_{i<j} a_{ijP} = 0$). Consequently, all connections in the estimated network $\widehat{A}_P$ are false positives. We use this to approximate the number of false positives in the estimated network. The approximation is

$$\sum_{(ij)\in\Theta} \hat{a}_{ij(\text{CI}(\gamma))} \approx \mathbb{E}\left[\sum_{i<j} \hat{a}_{ij(\text{CI}(\gamma))P}\right] = \widehat{\text{FP}}, \tag{26}$$

Scale–free network generated with huge R–package

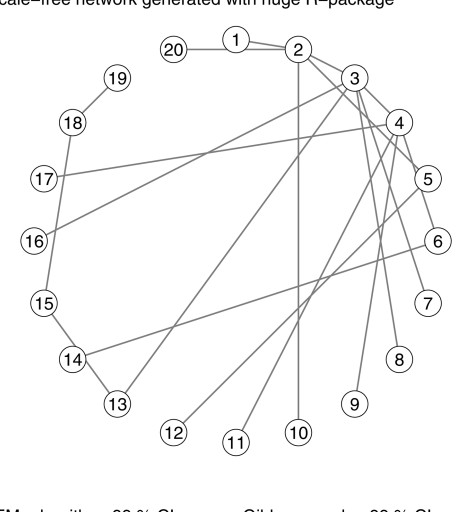

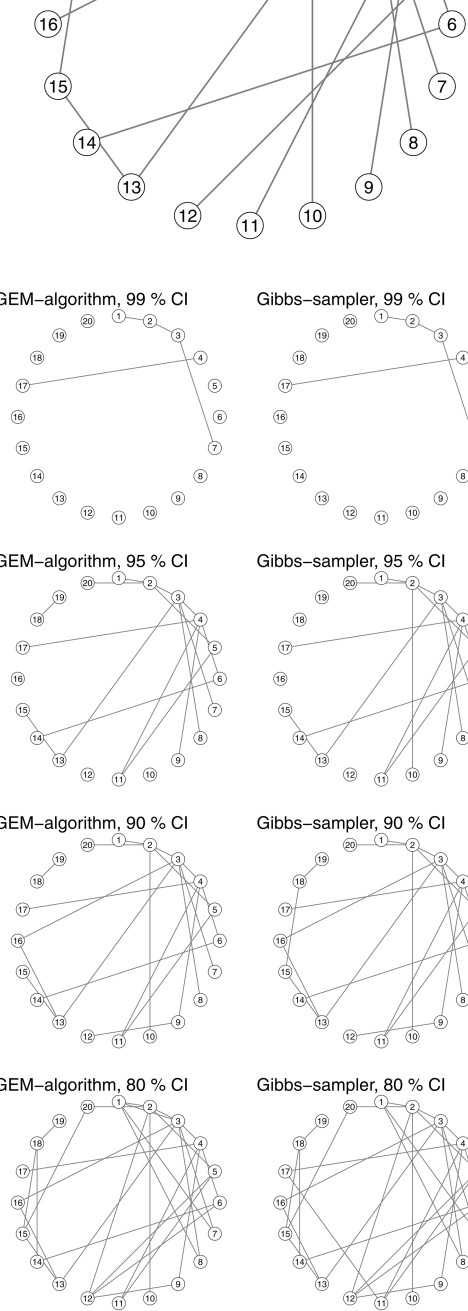

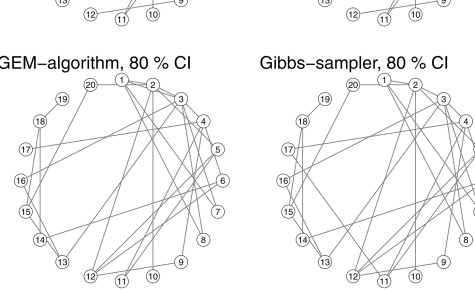

**Fig 4. Comparison of recovered networks with the Gibbs sampler and the GEM algorithm using credible intervals.** The graphical model considered in this example is a scale-free network generated with the R package `huge` ($n = 100, p = 20$).

where $\hat{a}_{ij(\mathrm{CI}(\gamma))P}$ is the adjacency matrix acquired with permuted data. The expected value $\mathbb{E}\left[\sum_{i<j} \hat{a}_{ij(\mathrm{CI}(\gamma))P}\right]$ is attained by repeating the permutation multiple times and calculating the median of $\sum_{i<j} \hat{a}_{ij(\mathrm{CI}(\gamma))P}$. For all results, we use 50 permutations. A similar permutation method for controlling FDR has been used with a fused lasso latent feature model [46], and permutations have also been used with network models [47,48]. The estimate of true positives is now $\widehat{\mathrm{TP}} = \sum_{i<j} \hat{a}_{ij(\mathrm{CI}(\gamma))} - \widehat{\mathrm{FP}}$.

We control FDR by adjusting the CI such that the target FDR is reached:

$$\widehat{\mathrm{CI}}(\gamma) = \underset{\mathrm{CI}(\gamma)}{\operatorname{argmin}}\left(\mathrm{FDR}_{\mathrm{target}} \geq \frac{\mathbb{E}\left[\sum_{i<j} \hat{a}_{ij(\mathrm{CI}(\gamma))P}\right]}{\sum_{i<j} \hat{a}_{ij(\mathrm{CI}(\gamma))}}\right), \tag{27}$$

where $\widehat{\mathrm{CI}}(\gamma)$ is the CI that produces the network with the specified target FDR. We illustrate the precision of this FDR control with multiple $\alpha$ values in Fig 5 and with different $\beta$ values in Figs H and I in S1 Text. The FDR control seems to be the most accurate with $\alpha$ values selected using the CC-method. The accuracy of FDR control also depends on the $\beta$ value, but a suitably selected $\alpha$ seems to be more important accuracy-wise.

## 2.9. An optimal credibility interval

With permutations, we get estimates for the number of false positives ($\widehat{\mathrm{FP}}$) and true positives ($\widehat{\mathrm{TP}}$). This enables us to use other metrics than FDR to select an optimal CI($\gamma$). In our assessment, maximizing the $F_1$-score will provide a graph that has an informative number of true positives compared to false positives [49]. An estimated $F_1$ is

$$\widehat{F}_1 = \frac{2 \cdot \widehat{\mathrm{TP}}}{2 \cdot \widehat{\mathrm{TP}} + \widehat{\mathrm{FP}} + \widehat{\mathrm{FN}}}, \tag{28}$$

where $\widehat{\mathrm{FN}} = (K - \widehat{\mathrm{TP}})$ and $K$ is the expected number of real connections in the network (see Section G in S1 Text). Our rule of thumb is to set it to the number of variables $p$, which we use for all results.

The optimal credible interval is achieved by maximizing the $\widehat{F}_1$:

$$\widehat{\mathrm{CI}}(\gamma) = \operatorname{argmax}_{\mathrm{CI}(\gamma)}(\widehat{F}_1). \tag{29}$$

We want to point out that the maximization metric can be changed from $F_1$ to some other value depending on the situation. For example, the Matthews Correlation Coefficient (MCC) can be used instead of $F_1$.

## 3. Results

### 3.1. Examples with simulated data

Simulated data enables us to compare our method to other network estimation methods. We opted to simulate our test data with two R packages, `huge` and `BDgraph`. In our experience, data simulated with `huge` is more difficult to estimate than `BDgraph`'s data in high-dimensional situations. We used both R packages to generate scale-free and cluster-type networks. Both packages generate the scale-free network structures in the same way, but the cluster structures differ. Both `huge` and `BDgraph` generate a network consisting of five disconnected clusters, meaning there are no connections between the clusters. The clusters are

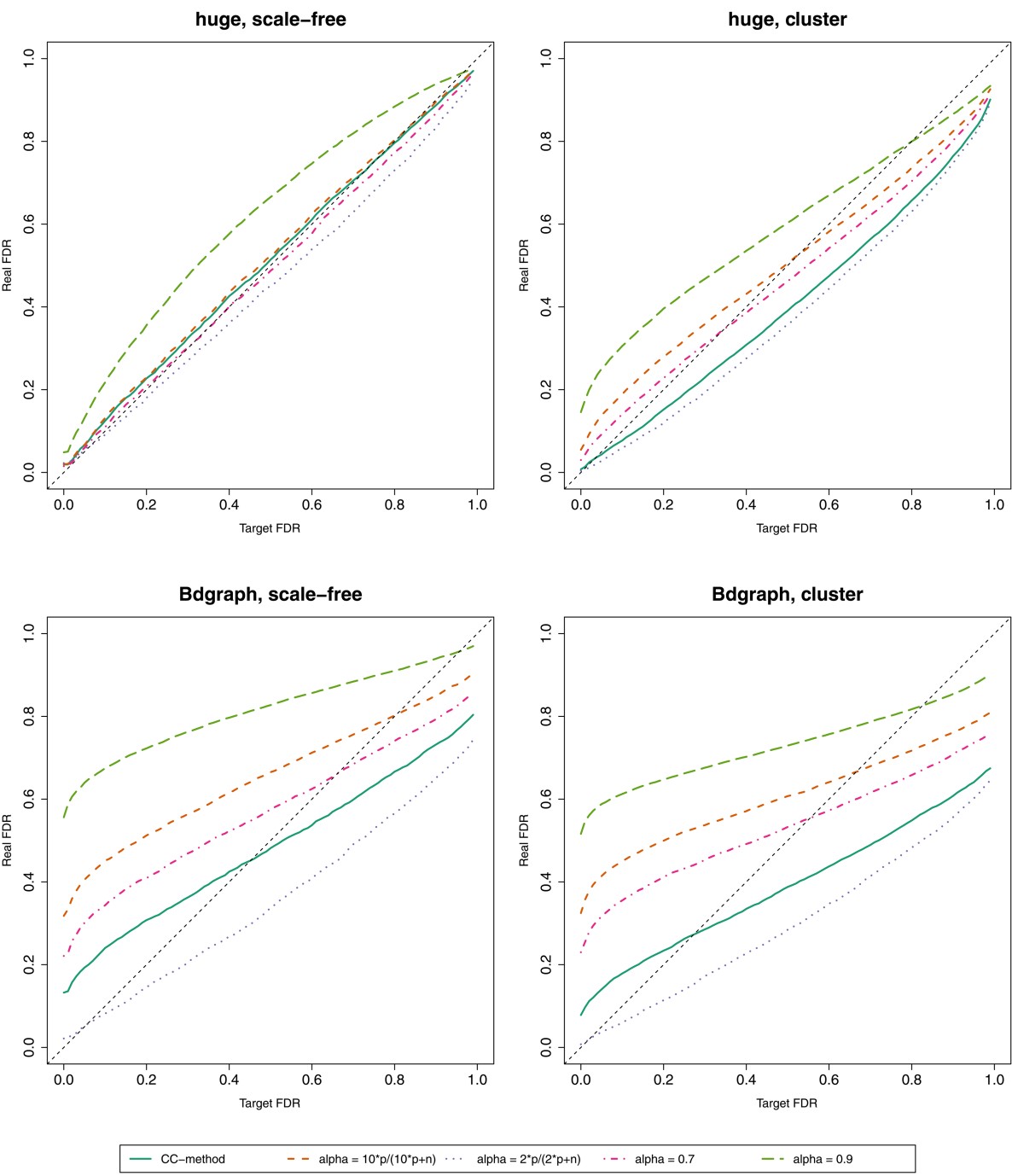

**Fig 5. Controlling FDR with multiple different $\alpha$ values.** Averages of 50 different simulated datasets. For all datasets, $p = 100$ and $n = 300$.

internally sparser with `BDgraph`, and the average clustering coefficient (ACC) is smaller with the network structure generated with `BDgraph`.

The main difference between `huge` and `BDgraph` is in how the data samples are generated and how the network structure is incorporated into the partial correlations. `huge` sets all partial correlation values to the same value that depends on the number of variables.

`BDgraph` randomly samples the partial correlations, and they can be anything between –1 and 1. To our knowledge, no established practice exists for simulating data with a specific network structure in mind. Thus, we are eager to use both data generation methods to illustrate how our method works well with both generators, which is not true for all GGM methods.

With both R packages and for both network structures, we generated datasets with sample sizes ($n$) of 35, 75, 150, and 300. For all sample sizes, the number of variables is 100 ($p$). With each sample size, we generated 50 different datasets. In Figs A-E in S1 Text, all these datasets were used.

For comparison we also picked six GGM methods: (i) graphical lasso (Glasso) [50] with stability approach for regularization selection (StARS) [51]; (ii) fast Bayesian GGM (FBGGM) [17]; (iii) G-Wishart using Birth-Death MCMC sampling algorithm [52]; (iv) CLEVEL [21, 53]; (v) thresholded adaptive validation (thAV) for Glasso [54]; (vi) tuning-insensitive graph estimation and regression (TIGER) [55]; and (vii) Bayesian Gaussian Graphical Models using matrix-F prior (BGGM) [24,56]. All of these have convenient R packages that make the comparison easy to implement. For Glasso with StARS, we used R packages `pulsar` [57] and `huge` [58], FBGGM is implemented with R package `beam` [17], G-Wishart with `BDgraph` [52], TIGER with R package `flare` [59], BGGM with R package `BGGM` [60], CLEVEL with R package `PCGII` [61], which also includes a new method called PCGII, that has the same properties than CLEVEL, but user can additionally incorporated a prior information to the estimation. BGGM implementation does not work with high-dimensional datasets and thus results with $n = 35, p = 100$ and $n = 75, p = 100$ are not available. BGGM is designed for use with psychological datasets that often have a large number of samples. For more details, see Section I in S1 Text.

We measured the network-recovery performance with four metrics for scale-free networks and three metrics for cluster networks. For scale-free, we report Matthews Correlation Coefficient (MCC), $F_1$-score ($F_1$), false discovery rate (FDR), and true positive rate (TPR). For the cluster network, we report the average clustering coefficient (ACC), normalized mutual information (NMI), and $F_1$-score. All metrics range from 0 to 1, except MCC, which has a value range from –1 to 1. For NMI, $F_1$, MCC, and TPR, values closer to 1 indicate better network recovery performance. For FDR, the lower value is considered preferable, and ACC is data-dependent. We considered $F_1$ the most informative metric for network estimation due to the highly uneven number of edges and absence of edges [49]. We described all metrics in greater detail in Section H in S1 Text.

**3.1.1. Computation time comparison between MCMC samplers and GEM algorithm.** In order to showcase the computational advantages of our GEM algorithm, we compared it with our implementation of Gibbs sampler, Gibbs sampler for matrix-F prior found in the R package `BGGM`, and Birth-Death MCMC sampler for G-Wishart prior in the R package `BDgraph`. In addition, we included computation times for our method with permutations for selecting an optimal credible interval and with the CC-method. We tested all methods with 100, 200, 300, and 400 variables. The test datasets were generated with the R package `BDgraph`. We repeated each run 5 times. The test computer uses an Intel Core i7-11700F processor with 32 GB of RAM. Computation time tests are heavily affected by the computer's capabilities, and we recognize the difficulty of reproducing these results. Regardless, this should still show the clear computational benefit of using the GEM algorithm over MCMC methods. The results are found in Table 2.

Computationally, our GEM algorithm is considerably faster than any MCMC method. Even with permutations and $\alpha$-selection using parallelization, our method outperforms the MCMC methods. Without parallelization, our method has computation times similar to

**Table 2. Comparisons of median computation times (in seconds) based on 5 runs for the GEM algorithm and the MCMC methods.** HMFGraph Gibbs, G-Wishart, and BGGM were run with 5000 iterations. For all $p$, $n = 2 \cdot p$.

| | Number of variables ($p$) | | | |
|---|---|---|---|---|
| | **100** | **200** | **300** | **400** |
| HMFGraph[1] | 0.3 | 1.4 | 5.5 | 14 |
| HMFGraph-P[2] | 3.9 | 12 | 38 | 91 |
| HMFGraph-P-$\alpha$[3] | 5.3 | 20 | 54 | 160 |
| HMFGraph-P-$\alpha$[4] | 12 | 71 | 246 | 680 |
| HMFGraph Gibbs | 14 | 91 | 305 | 730 |
| BGGM | 12 | 89 | 298 | 726 |
| G-Wishart | 17 | 90 | 489 | 1737 |

- [1] A single run of the GEM algorithm
- [2] The same as [1] with 50 permutation for selecting the optimal CI
- [3] The same as [2] but with $\alpha$ selected with the CC-method
- [4] The same as [3] but without parallelization

those of the Gibbs sampler and BGGM. Parallelizing our method is straightforward and helps computations immensely.

Memory usage is also a concern for larger networks. MCMC methods naturally use more memory because all samples of the precision matrix are saved. In R, this means an array with size of $p \times p \times$ (number of iterations). The GEM algorithm does not use a lot of memory because only two iterations have to be kept in memory simultaneously. Multiple GEM algorithms can be run in parallel without requiring a large amount of system memory.

**3.1.2. Simulated scale-free network datasets.** We present the results for datasets with a scale-free network structure in Table 3. Figs J-M in S1 Text illustrate how the recovered networks visually differ between all tested methods. With HMFGraph, we selected the best $\alpha$ value based on the CC-method (see Fig E in S1 Text) and the optimal credible interval by maximizing the estimated $F_1$-score. In addition, results with a target FDR of 0.2 are included. For all methods, default values were used, except for CLEVEL. A target FDR can be selected for CLEVEL, and we used the same 0.2 as with our method, but we also included the results with the default target FDR used in the R package (0.05). It should be noted that only CLEVEL and our HMFGraph have a built-in FDR control procedure. FDR control provides both HMFGraph and CLEVEL a benefit over all other methods. We can now also compare our FDR control procedure to CLEVEL's and find out how accurate the permutation-based approach is.

In general, our method performs well with all sample sizes and with both data generators when considering the $F_1$ and MCC values. In the worst case, our method has the third-best $F_1$ values with 75 and 150 samples with huge datasets. The other methods tested do not show this level of consistency with their network recovery capabilities. For example, Glasso with StARS has a rather competitive performance with huge datasets, but it has the second-worst performance with BDgraph datasets. The same can be observed with thAV, which works well with BDgraph, but not so well with huge. TIGER performs well with high-dimensional datasets, but the performance decreases relative to other methods when the sample size grows. In contrast, our method has either the best performance or close to the best with all sample sizes and with both generators.

The real FDR of our method is closer to the set target FDR than CLEVEL, especially with datasets generated with BDgraph. For instance, when $n = 75$, our method achieves, on average, an FDR value of 0.35 (target FDR = 0.2), but for CLEVEL, the corresponding FDR

**Table 3. Means of performance metrics.** The metrics are Matthews correlation coefficient (MCC), false discovery rate (FDR), $F_1$-score ($F_1$), and true positive rate (TPR). Averages of 50 different datasets. Standard deviations are in parentheses. The simulated networks have a scale-free structure and the datasets are generated with the R packages `huge` and `BDgraph`.

| Method | Data generation method | | | | | | | |
| --- | --- | --- | --- | --- | --- | --- | --- | --- |
| | `huge` | | | | `BDgraph` | | | |
| | MCC | FDR | $F_1$ | TPR | MCC | FDR | $F_1$ | TPR |
| **$n = 35, p = 100$** | | | | | | | | |
| HMFGraph (optimal CI) | 0.14 (0.05) | 0.85 (0.06) | 0.15 (0.05) | 0.20 (0.10) | 0.42 (0.08) | 0.47 (0.12) | **0.41** (0.08) | 0.35 (0.08) |
| HMFGraph $\left(\text{FDR}_{target} = 0.2\right)$ | 0.04 (0.06) | 0.45 (0.42) | 0.02 (0.03) | 0.01 (0.01) | **0.43** (0.08) | 0.35 (0.11) | 0.40 (0.10) | 0.31 (0.11) |
| Glasso StARS | **0.16** (0.05) | 0.84 (0.04) | **0.18** (0.05) | 0.20 (0.05) | 0.32 (0.05) | 0.77 (0.05) | 0.31 (0.05) | **0.51** (0.12) |
| FBGGM (`beam`) | 0.02 (0.05) | 0.22 (0.39) | 0.01 (0.01) | 0.00 (0.01) | 0.42 (0.07) | **0.34** (0.15) | 0.38 (0.10) | 0.30 (0.12) |
| G-Wishart | 0.08 (0.03) | 0.93 (0.02) | 0.10 (0.03) | 0.18 (0.05) | 0.20 (0.04) | 0.87 (0.02) | 0.20 (0.04) | 0.40 (0.09) |
| CLEVEL $\left(\text{FDR}_{target} = 0.2\right)$ | 0.03 (0.05) | 0.09 (0.23) | 0.01 (0.02) | 0.00 (0.01) | 0.33 (0.08) | 0.67 (0.21) | 0.31 (0.09) | 0.48 (0.19) |
| CLEVEL $\left(\text{FDR}_{target} = 0.05\right)$ | 0.01 (0.04) | **0.02** (0.14) | 0.00 (0.01) | 0.00 (0.00) | 0.35 (0.09) | 0.48 (0.27) | 0.32 (0.11) | 0.36 (0.19) |
| thAV | 0.12 (0.04) | 0.92 (0.03) | 0.12 (0.03) | **0.30** (0.10) | 0.40 (0.12) | 0.46 (0.21) | 0.39 (0.11) | 0.34 (0.08) |
| TIGER | **0.16** (0.06) | 0.74 (0.08) | 0.15 (0.05) | 0.11 (0.04) | 0.40 (0.07) | 0.59 (0.07) | **0.41** (0.07) | 0.43 (0.12) |
| BGGM | - | - | - | - | - | - | - | - |
| **$n = 75, p = 100$** | | | | | | | | |
| HMFGraph (optimal CI) | 0.32 (0.07) | 0.68 (0.10) | 0.33 (0.07) | 0.38 (0.09) | 0.53 (0.07) | 0.38 (0.10) | **0.53** (0.07) | 0.47 (0.08) |
| HMFGraph $\left(\text{FDR}_{target} = 0.2\right)$ | 0.26 (0.08) | 0.27 (0.14) | 0.18 (0.09) | 0.11 (0.06) | 0.53 (0.05) | 0.35 (0.09) | **0.53** (0.06) | 0.46 (0.10) |
| Glasso StARS | 0.33 (0.06) | 0.71 (0.05) | 0.34 (0.05) | 0.42 (0.07) | 0.38 (0.04) | 0.75 (0.06) | 0.35 (0.05) | **0.65** (0.11) |
| FBGGM (`beam`) | 0.18 (0.10) | 0.13 (0.16) | 0.09 (0.08) | 0.05 (0.05) | 0.52 (0.05) | 0.36 (0.12) | 0.52 (0.06) | 0.46 (0.10) |
| G-Wishart | 0.22 (0.05) | 0.80 (0.04) | 0.24 (0.05) | 0.29 (0.07) | 0.38 (0.05) | 0.70 (0.04) | 0.38 (0.05) | 0.53 (0.08) |
| CLEVEL $\left(\text{FDR}_{target} = 0.2\right)$ | 0.29 (0.09) | 0.27 (0.15) | 0.22 (0.10) | 0.14 (0.07) | 0.38 (0.10) | 0.69 (0.18) | 0.36 (0.12) | 0.64 (0.16) |
| CLEVEL $\left(\text{FDR}_{target} = 0.05\right)$ | 0.16 (0.09) | **0.06** (0.12) | 0.07 (0.06) | 0.04 (0.03) | 0.46 (0.10) | 0.52 (0.24) | 0.43 (0.12) | 0.55 (0.17) |
| thAV | 0.24 (0.04) | 0.87 (0.03) | 0.21 (0.04) | **0.56** (0.11) | **0.55** (0.07) | **0.15** (0.11) | 0.51 (0.08) | 0.37 (0.08) |
| TIGER | **0.35** (0.06) | 0.57 (0.07) | **0.36** (0.06) | 0.30 (0.06) | 0.49 (0.05) | 0.56 (0.06) | 0.49 (0.05) | 0.60 (0.12) |
| BGGM | - | - | - | - | - | - | - | - |
| **$n = 150, p = 100$** | | | | | | | | |
| HMFGraph (optimal CI) | 0.55 (0.08) | 0.43 (0.12) | 0.55 (0.08) | 0.55 (0.09) | **0.62** (0.05) | 0.29 (0.09) | **0.61** (0.06) | 0.55 (0.08) |
| HMFGraph $\left(\text{FDR}_{target} = 0.2\right)$ | 0.54 (0.09) | 0.23 (0.06) | 0.52 (0.11) | 0.40 (0.13) | 0.61 (0.05) | 0.33 (0.08) | **0.61** (0.05) | 0.58 (0.09) |
| Glasso StARS | 0.49 (0.06) | 0.64 (0.04) | 0.48 (0.06) | 0.71 (0.09) | 0.40 (0.04) | 0.75 (0.05) | 0.36 (0.06) | **0.74** (0.09) |
| FBGGM (`beam`) | 0.51 (0.10) | 0.14 (0.06) | 0.46 (0.12) | 0.32 (0.12) | 0.59 (0.05) | 0.36 (0.08) | 0.60 (0.05) | 0.57 (0.09) |
| G-Wishart | 0.48 (0.07) | 0.53 (0.06) | 0.49 (0.07) | 0.52 (0.09) | 0.54 (0.05) | 0.51 (0.06) | 0.55 (0.05) | 0.63 (0.08) |
| CLEVEL $\left(\text{FDR}_{target} = 0.2\right)$ | **0.58** (0.07) | 0.32 (0.07) | 0.57 (0.08) | 0.50 (0.11) | 0.45 (0.12) | 0.65 (0.19) | 0.42 (0.15) | 0.72 (0.13) |
| CLEVEL $\left(\text{FDR}_{target} = 0.05\right)$ | 0.54 (0.07) | **0.09** (0.06) | 0.48 (0.10) | 0.33 (0.09) | 0.55 (0.12) | 0.46 (0.25) | 0.53 (0.14) | 0.66 (0.14) |
| thAV | 0.38 (0.08) | 0.79 (0.08) | 0.33 (0.09) | **0.80** (0.08) | 0.60 (0.06) | **0.08** (0.06) | 0.56 (0.08) | 0.41 (0.09) |
| TIGER | 0.57 (0.07) | 0.44 (0.06) | **0.58** (0.07) | 0.60 (0.09) | 0.55 (0.04) | 0.56 (0.06) | 0.54 (0.04) | 0.71 (0.09) |
| BGGM | 0.15 (0.05) | 0.90 (0.03) | 0.15 (0.04) | 0.31 (0.07) | 0.24 (0.04) | 0.85 (0.03) | 0.23 (0.04) | 0.46 (0.06) |
| **$n = 300, p = 100$** | | | | | | | | |
| HMFGraph (optimal CI) | 0.78 (0.08) | 0.21 (0.10) | 0.78 (0.08) | 0.77 (0.09) | 0.69 (0.05) | 0.24 (0.08) | 0.69 (0.05) | 0.64 (0.06) |
| HMFGraph $\left(\text{FDR}_{target} = 0.2\right)$ | 0.77 (0.07) | 0.23 (0.05) | 0.77 (0.07) | 0.78 (0.12) | 0.68 (0.05) | 0.31 (0.07) | 0.68 (0.05) | 0.68 (0.06) |
| Glasso StARS | 0.75 (0.08) | 0.29 (0.07) | 0.75 (0.08) | 0.80 (0.11) | 0.49 (0.09) | 0.65 (0.13) | 0.46 (0.11) | 0.77 (0.08) |
| FBGGM (`beam`) | 0.78 (0.07) | 0.16 (0.05) | 0.78 (0.08) | 0.73 (0.12) | 0.66 (0.04) | 0.33 (0.05) | 0.66 (0.04) | 0.66 (0.06) |
| G-Wishart | 0.76 (0.06) | 0.26 (0.05) | 0.76 (0.06) | 0.79 (0.10) | **0.70** (0.05) | 0.31 (0.08) | **0.71** (0.05) | 0.73 (0.07) |
| CLEVEL $\left(\text{FDR}_{target} = 0.2\right)$ | 0.76 (0.07) | 0.29 (0.06) | 0.76 (0.06) | 0.83 (0.10) | 0.55 (0.14) | 0.56 (0.21) | 0.52 (0.17) | 0.79 (0.09) |
| CLEVEL $\left(\text{FDR}_{target} = 0.05\right)$ | **0.80** (0.07) | **0.08** (0.03) | **0.80** (0.08) | 0.71 (0.12) | 0.66 (0.13) | 0.36 (0.24) | 0.64 (0.15) | 0.74 (0.10) |
| thAV | 0.61 (0.13) | 0.57 (0.16) | 0.57 (0.15) | **0.93** (0.04) | 0.62 (0.06) | **0.06** (0.06) | 0.58 (0.08) | 0.43 (0.09) |
| TIGER | 0.73 (0.06) | 0.37 (0.04) | 0.72 (0.05) | 0.86 (0.08) | 0.59 (0.04) | 0.54 (0.07) | 0.58 (0.05) | **0.80** (0.07) |
| BGGM | 0.41 (0.06) | 0.76 (0.04) | 0.36 (0.05) | 0.76 (0.11) | 0.36 (0.03) | 0.79 (0.02) | 0.32 (0.03) | 0.67 (0.05) |

value is 0.69. This indicates that our permutation method is more flexible and can work with different data types.

BGGM has the worst network recovery capability out of all tested methods. This is perhaps due to that the implementation is designed for low-dimensional problems ($n \gg p$). As noted previously, the implementation in `BGGM` does not work with high-dimensional datasets

and thus only results with $n = 150$ and $n = 300$ are presented. Our method is noticeably better than BGGM when considering $F_1$ and MCC performance metrics, even though BGGM uses a matrix-F prior as our method, but without the hierarchical part. The results show that our hierarchical matrix-F prior and hyperparameter tuning is an immense improvement for high-dimensional datasets, and even when $n > p$.

**3.1.3. Simulated cluster network datasets.** For cluster network datasets, the results are found in Table 4. We ran our method with two different $\alpha$ values: the first results in the table are acquired with the CC-method, and the latter with $\alpha = 10 \cdot p/(10 \cdot p + n)$, which produces larger $\alpha$ values than the CC-method (see Fig E in S1 Text).

For our method, a large $\alpha$ value helps with clustering. NMI values are better for both data generators with larger $\alpha$ values. A major downside is that graph recovery capabilities suffer. For data generated with huge, the best $F_1$ values are still acquired with $\alpha = 10 \cdot p/(10 \cdot p + n)$, but for the data generated with BDgraph, the CC-method produces the best $F_1$ values, but NMI is considerably worse. The negative correlation between NMI and $F_1$ is true for other methods as well. For instance, Glasso with StARS has the best NMI for BDgraph dataset with $n = 300$, but its $F_1$ is the fourth lowest. Our method produces more clustered networks (larger ACC) with large $\alpha$ values than with $\alpha$ selected with the CC-method. $F_1$ values are better with our method when $\alpha$ is selected with the CC-method. This can be interpreted such that with large $\alpha$ values, the estimated networks are denser and less accurate than with $\alpha$ values selected with the CC-method, but the denser graph helps with identifying the clusters and cluster members.

As with scale-free networks, HMFGraph outperforms BGGM based on NMI and $F_1$ performance metrics. This further suggests that our method has superior network recovery capabilities in these situations.

## 3.2. Examples with real datasets

Simulated data rarely behaves in a similar way to a real dataset, and tests with real datasets can be considered necessary. We apply our method to two real datasets: a riboflavin dataset and an American gut dataset [62]. The riboflavin dataset is publicly available in the R package hdi [63], and the American gut dataset (*amgut2.filt.phy*) is available in the R package SpiecEasi [64]. With the riboflavin dataset, only 100 genes with the highest variance in expression were selected for the analysis, similar to [65] and [54]. The datasets have been tested with two other GGM methods before, namely with the Meinshausen and Bühlmann (MB) graph estimation method and with thAV [54,65,66]. This gives us an easy way to compare our results with those methods. As these datasets are not normally distributed, we first normalize both datasets with nonparanormal transformation using the R package huge [67].

**3.2.1. Riboflavin dataset.** The riboflavin dataset has 100 gene expression measurements and riboflavin production amounts collected from 71 observations ($p = 101$, $n = 71$). The study species is a bacterium called *bacillus subtilis*, from which gene expression and riboflavin production were measured. Riboflavin production was included as an extra node to the analysis. For our analysis, CI width was selected using permutations, and the target FDR was set to 0.2. The resulting graph is shown in Fig 6. For the same dataset, thAV produced a sparser graph than our method, but some similar structures can still be found [54]. The MB method recovered a denser graph in turn, which makes it much more difficult to structurally compare the graphs. They used StARS for parameter selection, which tends to produce dense graphs [51,65].

In Table 5, we listed all genes that are connected to the riboflavin production node, which is labeled as 1 in Fig 6 with varying target FDR values (see Fig N in S1 Text). When the target

**Table 4. Means of performance metrics.** The metrics are average clustering coefficient (ACC), normalized mutual information (NMI), and $F_1$-score ($F_1$). Averages of 50 different datasets. Standard deviations are in parentheses. The simulated networks have a cluster structure and the datasets are generated with the R packages `huge` and `BDgraph`.

| Method | Data generation method | | | | | |
| --- | --- | --- | --- | --- | --- | --- |
| | `huge` (ACC ≈ 0.303) | | | `BDgraph` (ACC ≈ 0.195) | | |
| | ACC | NMI | $F_1$ | ACC | NMI | $F_1$ |
| *n* = 35 | | | | | | |
| HMFGraph[1] (optimal CI) | 0.13 (0.05) | 0.26 (0.06) | 0.21 (0.05) | 0.43 (0.10) | 0.48 (0.07) | 0.42 (0.05) |
| HMFGraph[1] (FDR$_{target}$ = 0.2) | 0.52 (0.50) | 0.35 (0.01) | 0.03 (0.03) | 0.44 (0.09) | 0.49 (0.07) | 0.41 (0.06) |
| HMFGraph[2] (optimal CI) | 0.15 (0.06) | 0.26 (0.06) | 0.22 (0.05) | 0.53 (0.16) | 0.53 (0.11) | 0.42 (0.04) |
| HMFGraph[2] (FDR$_{target}$ = 0.2) | 0.49 (0.47) | **0.36** (0.01) | 0.04 (0.03) | 0.52 (0.13) | **0.56** (0.14) | 0.41 (0.05) |
| Glasso StARS | **0.16** (0.05) | 0.26 (0.05) | 0.22 (0.04) | 0.38 (0.11) | 0.53 (0.16) | 0.40 (0.03) |
| FBGGM (`beam`) | 0.84 (0.36) | 0.35 (0.01) | 0.01 (0.02) | 0.44 (0.09) | 0.48 (0.07) | 0.40 (0.07) |
| *G*-Wishart | 0.06 (0.01) | 0.14 (0.03) | 0.18 (0.03) | 0.08 (0.02) | 0.22 (0.06) | 0.28 (0.03) |
| CLEVEL (FDR$_{target}$ = 0.2) | 0.70 (0.42) | 0.35 (0.01) | 0.03 (0.04) | 0.48 (0.11) | 0.55 (0.13) | 0.38 (0.05) |
| CLEVEL (FDR$_{target}$ = 0.05) | 0.96 (0.20) | 0.35 (0.00) | 0.00 (0.01) | 0.57 (0.14) | 0.53 (0.13) | 0.37 (0.07) |
| thAV | 0.06 (0.02) | 0.18 (0.05) | **0.24** (0.03) | 0.18 (0.11) | 0.47 (0.09) | 0.43 (0.06) |
| TIGER | 0.11 (0.07) | 0.33 (0.02) | 0.15 (0.04) | **0.20** (0.05) | 0.50 (0.13) | **0.45** (0.06) |
| BGGM | - | - | - | - | - | - |
| *n* = 75 | | | | | | |
| HMFGraph[1] (optimal CI) | 0.12 (0.03) | 0.42 (0.07) | 0.38 (0.04) | 0.38 (0.09) | 0.57 (0.07) | 0.53 (0.04) |
| HMFGraph[1] (FDR$_{target}$ = 0.2) | 0.14 (0.12) | 0.41 (0.03) | 0.23 (0.07) | 0.37 (0.08) | 0.60 (0.09) | 0.54 (0.04) |
| HMFGraph[2] (optimal CI) | 0.17 (0.05) | 0.45 (0.07) | 0.39 (0.05) | 0.57 (0.13) | 0.67 (0.11) | 0.50 (0.04) |
| HMFGraph[2] (FDR$_{target}$ = 0.2) | 0.20 (0.08) | 0.44 (0.05) | 0.30 (0.08) | 0.53 (0.12) | **0.69** (0.14) | 0.48 (0.04) |
| Glasso StARS | 0.18 (0.04) | **0.46** (0.09) | **0.42** (0.04) | 0.38 (0.10) | 0.67 (0.14) | 0.45 (0.04) |
| FBGGM (`beam`) | 0.15 (0.24) | 0.38 (0.02) | 0.15 (0.07) | 0.35 (0.08) | 0.58 (0.08) | 0.53 (0.04) |
| *G*-Wishart | 0.04 (0.02) | 0.26 (0.06) | 0.32 (0.04) | 0.08 (0.02) | 0.39 (0.11) | 0.45 (0.04) |
| CLEVEL (FDR$_{target}$ = 0.2) | 0.23 (0.07) | **0.46** (0.07) | 0.36 (0.07) | 0.45 (0.11) | **0.69** (0.14) | 0.44 (0.07) |
| CLEVEL (FDR$_{target}$ = 0.05) | **0.25** (0.22) | 0.39 (0.03) | 0.16 (0.08) | 0.52 (0.14) | **0.69** (0.14) | 0.49 (0.06) |
| thAV | 0.10 (0.02) | 0.36 (0.09) | 0.40 (0.04) | 0.19 (0.11) | 0.54 (0.08) | 0.49 (0.06) |
| TIGER | 0.12 (0.04) | 0.44 (0.06) | 0.38 (0.05) | **0.20** (0.06) | 0.61 (0.13) | **0.55** (0.04) |
| BGGM | - | - | - | - | - | - |
| *n* = 150 | | | | | | |
| HMFGraph[1] (optimal CI) | 0.11 (0.03) | 0.72 (0.10) | 0.56 (0.05) | 0.31 (0.08) | 0.67 (0.07) | **0.63** (0.03) |
| HMFGraph[1] (FDR$_{target}$ = 0.2) | 0.10 (0.03) | 0.71 (0.12) | 0.55 (0.06) | 0.33 (0.07) | 0.71 (0.10) | **0.63** (0.03) |
| HMFGraph[2] (optimal CI) | 0.21 (0.05) | 0.79 (0.11) | 0.58 (0.04) | 0.55 (0.10) | 0.77 (0.10) | 0.56 (0.04) |
| HMFGraph[2] (FDR$_{target}$ = 0.2) | 0.21 (0.05) | 0.80 (0.12) | 0.59 (0.05) | 0.53 (0.10) | **0.80** (0.12) | 0.53 (0.05) |
| Glasso StARS | 0.23 (0.04) | 0.81 (0.10) | 0.58 (0.03) | 0.37 (0.09) | 0.77 (0.11) | 0.48 (0.05) |
| FBGGM (`beam`) | 0.10 (0.04) | 0.67 (0.11) | 0.52 (0.07) | 0.29 (0.05) | 0.67 (0.08) | **0.63** (0.03) |
| *G*-Wishart | 0.09 (0.03) | 0.62 (0.14) | 0.54 (0.05) | 0.09 (0.04) | 0.61 (0.11) | 0.61 (0.04) |
| CLEVEL (FDR$_{target}$ = 0.2) | **0.25** (0.04) | **0.82** (0.10) | 0.59 (0.03) | 0.40 (0.11) | 0.77 (0.11) | 0.50 (0.08) |
| CLEVEL (FDR$_{target}$ = 0.05) | 0.22 (0.06) | 0.67 (0.12) | 0.51 (0.07) | 0.43 (0.16) | 0.79 (0.13) | 0.57 (0.08) |
| thAV | 0.13 (0.02) | 0.77 (0.12) | 0.59 (0.05) | 0.18 (0.12) | 0.55 (0.08) | 0.52 (0.07) |
| TIGER | 0.15 (0.04) | 0.79 (0.10) | **0.60** (0.04) | **0.21** (0.04) | 0.71 (0.12) | 0.61 (0.02) |
| BGGM | 0.21 (0.03) | 0.17 (0.05) | 0.32 (0.04) | **0.21** (0.03) | 0.19 (0.05) | 0.33 (0.04) |
| *n* = 300 | | | | | | |
| HMFGraph[1] (optimal CI) | 0.14 (0.03) | 0.96 (0.04) | 0.79 (0.03) | 0.27 (0.07) | 0.78 (0.08) | 0.71 (0.03) |
| HMFGraph[1] (FDR$_{target}$ = 0.2) | 0.21 (0.04) | 0.97 (0.03) | 0.82 (0.03) | 0.30 (0.06) | 0.79 (0.09) | 0.71 (0.03) |
| HMFGraph[2] (optimal CI) | 0.24 (0.04) | 0.97 (0.04) | 0.78 (0.03) | 0.50 (0.07) | 0.84 (0.09) | 0.62 (0.03) |
| HMFGraph[2] (FDR$_{target}$ = 0.2) | 0.31 (0.04) | **0.98** (0.02) | 0.77 (0.02) | 0.49 (0.08) | **0.87** (0.09) | 0.59 (0.03) |
| Glasso StARS | **0.30** (0.04) | 0.97 (0.03) | 0.72 (0.02) | 0.52 (0.09) | **0.87** (0.08) | 0.55 (0.07) |
| FBGGM (`beam`) | 0.21 (0.04) | 0.97 (0.04) | 0.82 (0.03) | 0.24 (0.04) | 0.75 (0.07) | 0.70 (0.03) |
| *G*-Wishart | 0.19 (0.03) | 0.96 (0.04) | **0.83** (0.04) | 0.13 (0.04) | 0.77 (0.08) | **0.74** (0.03) |
| CLEVEL (FDR$_{target}$ = 0.2) | 0.28 (0.03) | 0.97 (0.03) | 0.72 (0.02) | 0.32 (0.12) | 0.81 (0.11) | 0.57 (0.09) |
| CLEVEL (FDR$_{target}$ = 0.05) | 0.22 (0.03) | **0.98** (0.03) | 0.76 (0.03) | 0.34 (0.15) | 0.85 (0.09) | 0.65 (0.09) |
| thAV | 0.21 (0.03) | 0.97 (0.03) | 0.79 (0.03) | 0.16 (0.10) | 0.57 (0.08) | 0.54 (0.07) |
| TIGER | 0.24 (0.03) | 0.97 (0.02) | 0.79 (0.02) | **0.21** (0.05) | 0.78 (0.11) | 0.66 (0.02) |
| BGGM | 0.18 (0.01) | 0.69 (0.12) | 0.64 (0.03) | 0.16 (0.02) | 0.27 (0.06) | 0.47 (0.03) |

- [1] HMFGraph with $\alpha$ selected using the CC-method
- [2] HMFGraph with $\alpha = 10 \cdot p / (10 \cdot p + n)$

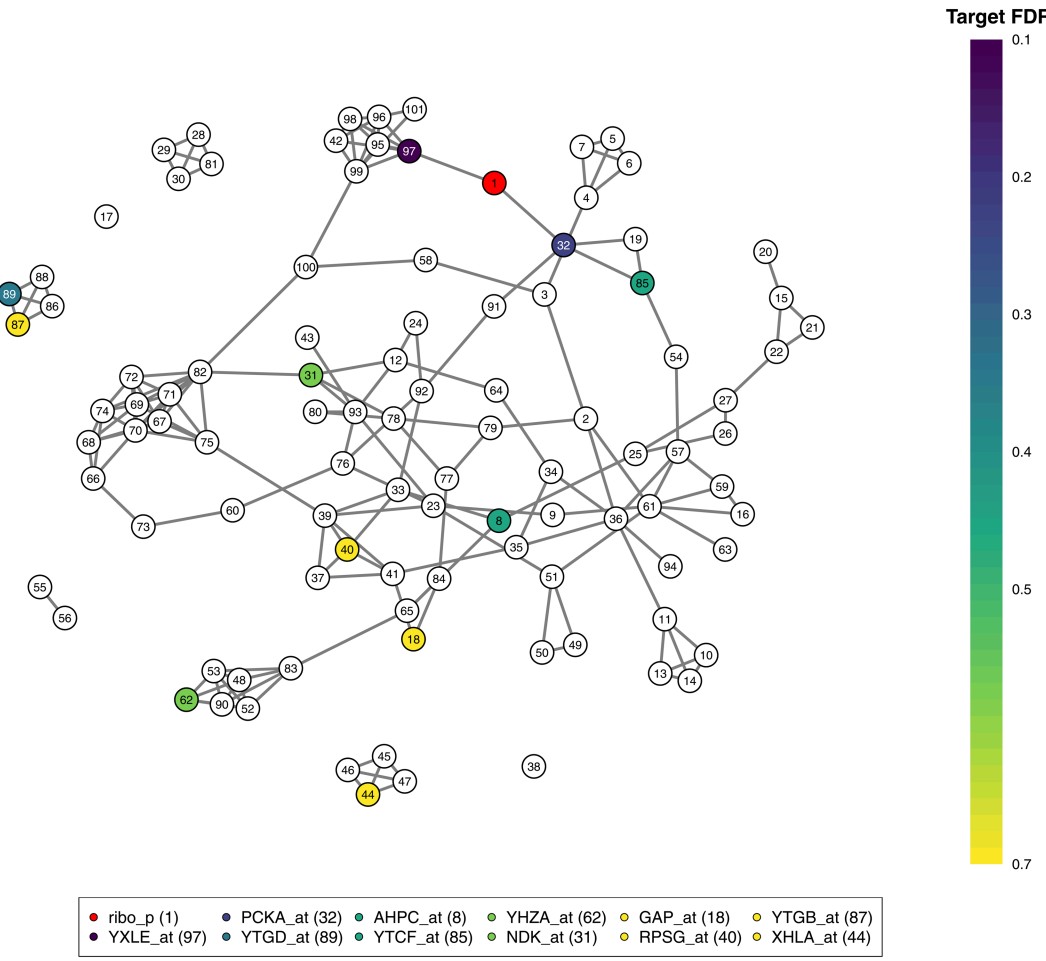

**Fig 6. The network recovered from the riboflavin dataset using the HMFGraph method.** The network was constructed with a target FDR set to 0.2. The node describing the riboflavin production is highlighted in red. Genes linked to riboflavin production are highlighted with colors describing the target FDR level, triggering a connection.

**Table 5**. **Genes connected to the riboflavin production node with different target FDR levels.**

| Target FDR level | Genes connected to riboflavin production |
|---|---|
| 0 | - |
| 0.1 | YXLE_at |
| 0.2 | Same as above + PCKA_at |
| 0.3 | Same as above + YTGD_at |
| 0.4 | Same as above + AHPC_at, YTCF_at |
| 0.5 | Same as above + NDK_at, YHZA_at |
| 0.6 | Same as above |
| 0.7 | Same as above + GAP_at, RPSG_at, YTGB_at, XHLA_at |

FDR increased, the number of connections to ribolavin production increased. Interestingly, almost all listed genes have been previously linked to riboflavin production in earlier analyses with this dataset [63,65,68–70]. The first gene that connects to the riboflavin production is YXLE_at. The same gene was found to be linked to riboflavin production with several multiple regression methods [68,70]. The other genes are less commonly found by other

methods, and those that require a larger target FDR value to be recovered are more likely to be false positives or falsely identified partial correlations.

An advantage of using the GGM method is that one can visualize all gene co-expressions in addition to the genes connected with the riboflavin production. For example, from the graph we can see that YXLE_at is a part of a larger cluster (including nodes 42, 95-99 and 101, see Section K in S1 Text). This cluster includes a gene YXLD_at (node 96), which is often linked to riboflavin production [65,68,69,71]. Based on our analysis, we can postulate that the gene YXLD_at is not directly linked or partially correlated with riboflavin production, but the apparent connection is due to the gene YXLE_at. Also, the whole cluster has been identified previously (see STRING database [72,73]) and many genes in it have been shown to interact biologically with each other (e.g., for YXLE_at and YXLD_at [74]).

**3.2.2. American gut dataset.** The gut dataset consists of 296 samples of 138 operational taxonomic units (OTUs). In addition, each OTU's respective taxon is known. We used two methods for selecting suitable $\alpha$ values. Fig 7A illustrates the network recovered with HMF-Graph, whose $\alpha$ value was selected with the CC-method. Furthermore, we used a larger $\alpha$ value ($\alpha = 10 \cdot p/(10 \cdot p + n) \approx 0.823$) in Fig 7B. We selected the optimal credible interval for both $\alpha$ values based on the estimated $F_1$-score. With a larger $\alpha$ value, the recovered graph has a more clustered structure. The obtained network clusters closely follow the biological taxon level *order*. This behavior is similar to the one observed with the simulated datasets. Based on the remarks with the cluster dataset results, we can postulate that the network recovered

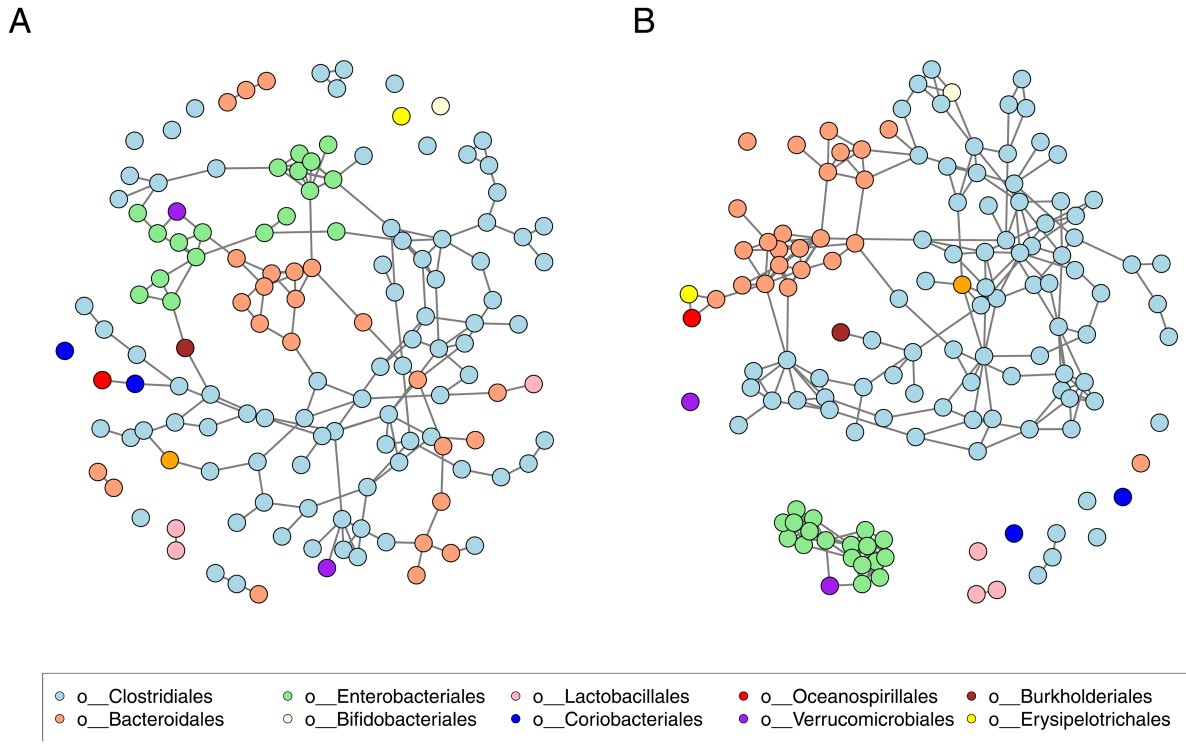

**Fig 7. The network recovered from the gut dataset using HMFGraph.** An optimal credible interval was selected using 50 permutations. Different OTU groups (in *order* taxonomic rank) are illustrated with different colors. **(A)** An optimal value of parameter $\alpha$ was selected based on the CC-method ($\alpha \approx 0.39$). **(B)** The value of parameter $\alpha$ was set to be large (e.g., $\alpha = 10 \cdot p/(10 \cdot p + n) \approx 0.82$).

with an optimal $\alpha$ value is closer to the real partial correlation structure in Fig 7A, but the underlying clusters are easier to recognize in Fig 7B, where a larger $\alpha$ value is used.

The found clusters in Fig 7B are similar to those recovered with thAV [54]. For example, OTUs from *Enterobacteriales order* are closely grouped together, i.e., they have a lot of connections between each other in the same *order*, but almost no connections to OTUs in other *orders*. We additionally noticed that the clustering behavior between different taxa levels has been observed before. For instance, in [6], different OTUs on the taxon level *family* were found to have clustering properties. For comparison, we added our networks in Fig 7 with a different coloring scheme to highlight the *family* taxon (see Fig O in S1 Text). We can observe that with our analysis, we also see that OTUs from different taxa *familiae* do not partially correlate. A good example is *familiae Bifidobacteriaceae* and *Bacteroidaceae*, which do not have any connection between them, as was also observed in [6]. This can be interpreted such that OTUs from *familiae Bifidobacteriaceae* and *Bacteroidaceae* are found independently of each other in the human gut.

Generally, on *phylum* taxonomic level, the found network clusters do not follow *phyla* groups as closely as on other taxa levels [75–78]. For instance, in [77], where they analyzed another gut microbiome dataset, the clustering between taxa *phyla* was noticeable, but there were connections between OTUs in different *phyla*. In Fig P in S1 Text, *phyla* taxa are highlighted with different colors in our recovered network. The results show that our method clusters the same *phyla* as in [77], but there are fewer connections between the clusters. A similar phenomenon was observed with other human microbiome datasets [76,78], where found clusters were similar to *phyla* groups. Our results can be interpreted as follows: if one OTU is found from some *phylum*, then it is more likely that other OTUs from the same *phylum* are also present.

## 4. Discussion

In this article, we introduced a novel hierarchical matrix-F prior for GGMs and showed its capabilities with high-dimensional datasets. The results demonstrated that a model using our hierarchical matrix-F prior is able to recover network structures in most cases better, or at least similarly, than competing methods. In addition, we provided a fast GEM algorithm with computational advantages over previous MCMC methods.

Many biological data do not fully follow the Gaussian assumption and also exhibit heterogeneity, such as the single-cell gene expressions, and may also be zero-inflated and compositional in nature (e.g., microbiome datasets). However, our method and often other GGMs are based on the assumption that the analyzed data are normally distributed, exceptions being, e.g., BLGGM [8]. In example analysis, we followed [54] and handled this problem by using a nonparanormal transformation [67]. Our analysis results illustrated that this approach can produce meaningful networks with non-Gaussian datasets. If a case study of the specific type of observations (e.g., zero-inflated microbiome data) is conducted, then a more thorough consideration of the validity of the data normalization procedure and model assumptions is recommended.

In our view, there is still work to be done in selecting the correct values for the parameters $\alpha$ and $\beta$ (or $\nu$ and $\delta$). Thanks to our reparameterization, interpretations for $\alpha$ and $\beta$ are now easier and more intuitive than with $\nu$ and $\delta$. We also explained a suitable value range of $\alpha$ when dealing with high-dimensional datasets and showed that the CC-method based on

Ledoit-Wolf covariance shrinkage seems to work adequately. More detailed research is still required for the parameter selection.

We believe the optimal credible interval selection based on permutation can be used flexibly. Here, we used the credible interval that maximized the estimated $F_1$-score, but other criteria can also be considered. Depending on the situation, this may yield a more informative graph recovery.

We want to expand our method for dynamic Gaussian graphical models in the future. A possible way of introducing a time dependency for this model is by setting the *B* matrix in the prior (10) to be a precision matrix of a previous time point.

## Supporting information

**S1 Text. Supporting information document.** Includes all supporting information materials. (PDF)

## Author contributions

**Conceptualization:** Aapo E. Korhonen, Olli Sarala, Tuomas Hautamäki, Markku Kuismin, Mikko J. Sillanpää.

**Data curation:** Aapo E. Korhonen.

**Formal analysis:** Aapo E. Korhonen.

**Funding acquisition:** Mikko J. Sillanpää.

**Investigation:** Aapo E. Korhonen.

**Methodology:** Aapo E. Korhonen, Olli Sarala, Tuomas Hautamäki, Markku Kuismin, Mikko J. Sillanpää.

**Project administration:** Mikko J. Sillanpää.

**Resources:** Aapo E. Korhonen, Mikko J. Sillanpää.

**Software:** Aapo E. Korhonen, Tuomas Hautamäki.

**Supervision:** Markku Kuismin, Mikko J. Sillanpää.

**Validation:** Aapo E. Korhonen, Olli Sarala, Tuomas Hautamäki.

**Visualization:** Aapo E. Korhonen, Olli Sarala, Tuomas Hautamäki.

**Writing – original draft:** Aapo E. Korhonen.

**Writing – review & editing:** Aapo E. Korhonen, Olli Sarala, Tuomas Hautamäki, Markku Kuismin, Mikko J. Sillanpää.

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
