## [Decision Letter · Decision Letter 0]

25 Jun 2025

PCOMPBIOL-D-25-00977

HMFGraph: Novel Bayesian approach for recovering biological networks

PLOS Computational Biology

Dear Dr. Korhonen,

Thank you for submitting your manuscript to PLOS Computational Biology. After careful consideration, we feel that it has merit but does not fully meet PLOS Computational Biology's publication criteria as it currently stands. Therefore, we invite you to submit a revised version of the manuscript that addresses the points raised during the review process.

Please submit your revised manuscript within 60 days Aug 25 2025 11:59PM. If you will need more time than this to complete your revisions, please reply to this message or contact the journal office at ploscompbiol@plos.org. Please include the following items when submitting your revised manuscript:

We look forward to receiving your revised manuscript.

Kind regards,

Lun Hu

Academic Editor

PLOS Computational Biology

Mark Alber

Section Editor

PLOS Computational Biology

**Journal Requirements:**

2) Your manuscript is missing the following section: Results.  Please ensure all required sections are present and in the correct order. Make sure section heading levels are clearly indicated in the manuscript text, and limit sub-sections to 3 heading levels. An outline of the required sections can be consulted in our submission guidelines here:

4) Please provide a completed 'Competing Interests' statement, including any COIs declared by your co-authors. If you have no competing interests to declare, please state "The authors have declared that no competing interests exist". 

**Reviewers' comments:**

Reviewer's Responses to Questions

Reviewer #1: The manuscript introduces a fast Bayesian Gaussian Graphical Model using a hierarchical matrix-F prior, offering competitive network recovery and efficient hyperparameter tuning via condition number constraints. It further proposes a novel edge selection method based on credible intervals optimized by F1-score, with superior computational performance over MCMC-based approaches.

The authors have conducted substantial work in the field of recovering biological networks. However, my main concerns are as follows:

1.The abstract should clearly articulate the current limitations in existing approaches in order to highlight the novelty and contributions of this work.

2.The introduction is recommended to be enriched with more background on real-world biological applications of network recovery, such as cancer pathway inference or metabolic network reconstruction, in order to better demonstrate the practical relevance and appeal of the method.

3.While the proposed hierarchical Matrix-F prior is theoretically innovative, a more systematic and quantitative comparison with existing Matrix-F-based models is needed—particularly regarding stability and inference performance in high-dimensional settings.

4.The section on Gaussian graphical models reads more like a preliminaries/background section than a direct part of the model description. It might be clearer if repositioned or retitled accordingly.

5.The methodological presentation requires a major revision. Rather than listing only what modifications were made to the model, I would prefer a more structured exposition that traces the complete modeling pipeline—from input to output—with the innovations clearly embedded within that logical flow.

6.A high-level schematic or workflow diagram summarizing the modeling procedure and its innovations would significantly improve clarity and accessibility.

7.The manuscript should more clearly explain the differences between the generalized expectation-maximization (GEM) algorithm and the standard EM algorithm, and elaborate on why GEM is particularly effective in the context of this model.

8.The performance comparisons across methods rely on fixed α and CI strategies for the proposed model. To ensure fairness, I suggest applying a consistent criterion (e.g., a fixed FDR threshold such as 0.2) across all methods.

9.Although real biological datasets are analyzed, the interpretations remain largely qualitative. Incorporating more biological validation—such as comparison with known gene interactions or pathway annotations—would greatly strengthen the biological significance of the recovered networks.

10.Given the large number of mathematical symbols (e.g., α, β, ν, δ), I recommend including a notation table in the methods section to improve readability and reduce potential confusion.

Reviewer #2: This paper proposes a new way to construct biological networks from biological data using Bayesian Gaussian graphical models. Its contributions include incorporating a matrix-F prior, developing a fast generalized EM algorithm, and designing a scheme to select hyperparameters. The methodology part of this paper is statistically sound. However, I have some comments that need more explanations by the authors.

1. The introduction section of this paper is too statistically technical. As this paper is motivated by constructing biological networks, it would be more interesting to general audiences if the authors can make clear which kind of biological network you want to recover, protein network, gene network, or gut network, and introduce some neccessary biological backgrounds.

2. My second question is about the validity of the Gaussian assumption. Many biological data do not fully follow the Gaussian assumption and also exhibit heterogeneity, such as the single cell gene expressions. In the real applications, the paper analyzes gene expression data and gut data. Therefore, it is important to verify the Gaussian assumption in this data. In my opinion, one important statistical paper to address single cell gene expression data is “Estimating heterogeneous gene regulatory networks from zero-inflated single-cell expression data”, The Annals of Applied Statistics, 2022. The authors may emphasize the difference between the proposed method HMFGraph and this AOAS paper, and I believe this AOAS paper can further motivate the improvement of HMFGraph when applied to zero-inflated and heterogeneous biological data.

3. Regarding the edge selection in section 2.5, why do not the authors apply a variable selection prior to the edges, such spike-slab prior or horseshoe prior? The authors should provide some explanations.

4. About the hyperparameter selection, users should calculate Cond(\hat{\Omega}_{\alpha}) for each alpha. What is the computational time for one single alpha? Is it practically applicable? Moreover, could the authors implement some sensitivity analyses for these hyperparameters? In this way, we can largely reduce the selection range of the hyperparameters.

5. The explations for the results in real datasets are shallow. Could the authors provide more biological insights from the results by HMFGraph? More biological findings rather than simulation results are more interesting to Plos computational biology readers.

Reviewer #3: The authors introduce HMFGraph, a Bayesian GGM method that employs a hierarchical matrix-F prior, which computationally outperforms existing Bayesian GGM approaches in handling high-dimensional biological data. The use of this prior enhances flexibility and adaptability to different network structures. The manuscript was well-written and easy to read. My concerns are as follows.

1.Justification of the introduced new parameter \beta :

Figure 3 showed the curves of FDR with varying \alpha, however, the role and selection process for \beta require further explanation or empirical validation to ensure readers understand its impact on network recovery.

2.Sensitivity/Specificity Analysis:

The authors reported the FDR of varying \alpha. Reporting sensitivity/specificity for both \alpha and \beta would help guide users in parameter selection.

3. Biological Interpretation:

The biological significance of the results, particularly in the gut microbiome study, could be strengthened with a comparison to results in the literature.

4.page 5/23, line 135, \beta \in [0,1[should be changed to \beta \in [0,1]

**Have the authors made all data and (if applicable) computational code underlying the findings in their manuscript fully available?**

Reviewer #1: Yes

Reviewer #2: Yes

Reviewer #3: Yes

PLOS authors have the option to publish the peer review history of their article (what does this mean?). If published, this will include your full peer review and any attached files.

Reviewer #1: No

Reviewer #2: No

Reviewer #3: No

**Figure resubmission:**
---

## [Decision Letter · Decision Letter 1]

1 Oct 2025

PCOMPBIOL-D-25-00977R1

HMFGraph: Novel Bayesian approach for recovering biological networks

PLOS Computational Biology

Dear Dr. Korhonen,

Thank you for submitting your manuscript to PLOS Computational Biology. After careful consideration, we feel that it has merit but does not fully meet PLOS Computational Biology's publication criteria as it currently stands. Therefore, we invite you to submit a revised version of the manuscript that addresses the points raised during the review process.

Please submit your revised manuscript within 30 days Dec 01 2025 11:59PM. If you will need more time than this to complete your revisions, please reply to this message or contact the journal office at ploscompbiol@plos.org. Please include the following items when submitting your revised manuscript:

We look forward to receiving your revised manuscript.

Kind regards,

Lun Hu

Academic Editor

PLOS Computational Biology

Marc Birtwistle

Section Editor

PLOS Computational Biology

**Additional Editor Comments :**

One of our reviewers still has some minor concerns regarding this work, and authors are suggested to carefully address these issues.

**Reviewers' comments:**

Reviewer's Responses to Questions

Reviewer #1: The authors have addressed most of the issues; however, I still have two minor comments that I would like them to revise:

1.The flowchart is rather simplistic. Please enhance it with more relevant illustrations for better visualization and also further optimize the layout and spacing of the figures.

2.In light of the authors’ contributions to biological networks, I suggest that the authors consider citing the following article at appropriate places in the manuscript.

DOI:10.1038/s41551-024-01312-5

DOI: 10.1109/JBHI.2025.3585290

Reviewer #2: I thank the authors for addressing my concerns well.

Reviewer #3: The authors have addressed all my concerns, I have no further comments.

**Have the authors made all data and (if applicable) computational code underlying the findings in their manuscript fully available?**

Reviewer #1: None

Reviewer #2: None

Reviewer #3: Yes

PLOS authors have the option to publish the peer review history of their article (what does this mean?). If published, this will include your full peer review and any attached files.

Reviewer #1: No

Reviewer #2: No

Reviewer #3: No

**Figure resubmission:**
---

## [Decision Letter · Decision Letter 2]

14 Oct 2025

Dear Mr. Korhonen,

We are pleased to inform you that your manuscript 'HMFGraph: Novel Bayesian approach for recovering biological networks' has been provisionally accepted for publication in PLOS Computational Biology.

Best regards,

Lun Hu

Academic Editor

PLOS Computational Biology

Marc Birtwistle

Section Editor

PLOS Computational Biology

All reviewers were satisfied with the changes made in this revision, and they have no further comments.

Reviewer's Responses to Questions

**Comments to the Authors:**

Reviewer #1: The author answered my opinion very well. I have no more opinions.

**Have the authors made all data and (if applicable) computational code underlying the findings in their manuscript fully available?**

Reviewer #1: Yes

PLOS authors have the option to publish the peer review history of their article (what does this mean?). If published, this will include your full peer review and any attached files.

Reviewer #1: No

---

## [Editor Report · Acceptance letter]

PCOMPBIOL-D-25-00977R2

HMFGraph: Novel Bayesian approach for recovering biological networks

Dear Dr Korhonen,

I am pleased to inform you that your manuscript has been formally accepted for publication in PLOS Computational Biology. Your manuscript is now with our production department and you will be notified of the publication date in due course.

With kind regards,

Olena Szabo
